# Assimilating WIVERN winds pseudo-observations in WRF model: an application to the outstanding case of the Medicane Ianos

Stefano Federico<sup>1</sup>, Rosa Claudia Torcasio<sup>1</sup>, Claudio Transerici<sup>1</sup>, Mario Montopoli<sup>1</sup>, Cinzia Cambiotti<sup>2</sup>, Francesco Manconi<sup>2</sup>, Alessandro Battaglia<sup>2</sup>, and Maryam Pourshamsi<sup>3</sup>

<sup>1</sup>CNR-ISAC, via del Fosso del Cavaliere 100, 00133 Rome

<sup>2</sup>Dipartimento di Ingegneria dell'Ambiente, del Territorio, Politecnico di Torino, Turin, Italy

<sup>3</sup>ESA-ESTEC, Noordwijk, Netherlands

**Abstract.** Accurate weather forecasts are important to our daily lives. Wind, cloud and precipitation are key drivers of the Earth's water and energy cycles, and they can also pose weather-related threats, making the task of numerical weather prediction (NWP) models particularly challenging and important.

~~The Wind Velocity Radar Nephoscope (WIVERN) mission will be the first space-based mission to provide global in-cloud wind measurements, and also the first to deliver simultaneous observations of winds, clouds and precipitation. The mission is proposed as a candidate for the European Space Agency (ESA)'s Earth Explorer 11 within the Future Earth Observation (FutureEO) programme. It is currently in Phase A, with the recommendation decision expected in July 2025. If~~

The Wind Velocity Radar Nephoscope (WIVERN) mission will be the first space-based mission to provide global vertical profiles of winds within clouds, and to deliver simultaneous observations of winds, clouds, and precipitation with unprecedented resolution and coverage. The mission has been selected as the European Space Agency's (ESA) Earth Explorer 11 within the Future Earth Observation (FutureEO) programme. Its data could be beneficial to several sectors: improving our knowledge of weather phenomena, validate climate statistics, and enhancing NWP performance. This paper aims to contribute to the last point by analyzing the impact ~~that WIVERN would have in the case of a Tropical-like cyclone (TLC) event. In this work, The impact~~ of assimilating WIVERN Line of Sight (LoS) winds ~~, retrieved from WIVERN Doppler measurements,~~ on NWP performance ~~is assessed~~, for the high-impact case study of Medicane Ianos, which occurred between 15 and 21 September 2020 in the central Mediterranean and made landfall on the west coast of Greece.

To this end, we generate WIVERN pseudo-observations, that are assimilated in the Weather Research and Forecasting (WRF) model run at moderate horizontal resolution (4 km).

Results show that assimilating WIVERN into the WRF model has a positive impact on the prediction of the Medicane trajectory. Specifically, assimilating WIVERN just once ~~improves reduces~~ the trajectory forecast error by ~~about 40%~~. The data assimilation of WIVERN pseudo-observations affects not only the storm's trajectory but also its physical characteristics. It is also shown that the assimilation improves the prediction of precipitation and surface winds, and has the potential to improve our resilience to severe weather events by enabling better forecasts of storm impacts. Finally, we present the results of ~~two~~ ~~three~~ sensitivity experiments in which the background and observation errors were ~~changed different~~. The results show greater sensitivity to changes in the background error matrix.

## 1 Introduction

Numerical weather prediction (NWP) is an initial and boundary condition problem, and its performance depends, among other factors, on an accurate representation of the initial atmospheric state. The purpose of data assimilation (DA, Kalnay et al. (2024)) is to find a model state that provides the best match between the most recent model prediction and the available obser-

vations. This model state, called the analysis, can then be used to start a new model forecast with improved initial conditions.

In-cloud winds and, more in general, winds are key to understanding cloud and storm processes and the coupling between water, heat, and atmospheric circulation, yet they remain one of the major gaps in the Global Observing System (GOS) (Baker et al., 2014). Indeed, wind observations are sparse and unevenly distributed, especially in cloudy regions, on the mesoscale, and in the vertical dimension. Given the importance of wind observations, the following, two specific recommendations were made

to Space Agencies at the 7th World Meteorological Organisation (WMO) workshop on “The impact of Various Observing Systems on NWP” (WMO, 2020): a) Space Agencies are encouraged to continue pursuing wind profile measurements from space; b) Effort is encouraged to assess complementary/synergies between different wind measurement systems/technologies (e.g. Aeolus and Atmospheric Motion Vectors). As winds are lacking observations, usually they show an important impact on the NWP forecast when assimilated (Horányi et al., 2015; Li et al., 2023; Rennie et al., 2021).

Between 2018-2023, the European Space Agency (ESA)’s Aeolus satellite provided wind observations along the Horizontal Line of Sight (HLoS) through atmospheric columns in optically thin clouds and clear sky, using a Doppler Wind Lidar (Illingworth et al., 2015). The positive impact of Aeolus data assimilation in NWP models has been demonstrated by major meteorological centers (Rani et al., 2022; Garrett et al., 2022; Martin et al., 2023; Rennie et al., 2021). Aeolus observations have also been assimilated in limited-area NWP models, showing, albeit to a lesser extent compared to global models, a positive impact (Stathopoulos et al., 2023; Matsangouras et al., 2023; Hagelin et al., 2021; Feng and Pu, 2023).

The WInd VElocity Radar Nephoscope (WIVERN) (Illingworth et al. (2018); Battaglia et al. (2022); Tridon et al. (2023)) has been selected by ESA recommended to be launched for the 11th Earth Explorer mission. ~~one of the two candidates~~ Earth Explorer missions are proposed by the scientific community to demonstrate how breakthrough technology can deliver an astounding range of scientific findings about our planet. As reported in the ESA Earth Explorer missions web page (https://www.esa.int/Enabling\_Support/Operations/Earth\_Explorers), these missions focus on the atmosphere, biosphere, hydrosphere, cryosphere and Earth’s interior with an overall emphasis on learning more about the interactions between these systems and the impact that human activity is having on Earth’s natural processes. ~~The mission is successfully selected for implementation~~ WIVERN, will carry a 94 GHz radar with a conically scanning 800 km swath. It will provide, for the first time, profiles of in-cloud winds and precipitation on a global scale, with a revisit time of 1.5 days at the equator. The implications of this unprecedented sampling capability have been recently discussed in Tridon et al. (2023) and Scarsi et al. (2024a). WIVERN measurements will have a vertical resolution of approximately 600 m and an instantaneous footprint of less than  $1 \times 1 \text{ km}^2$  (Illingworth et al., 2018). Wind measurements will then be averaged along the scanning direction over 5 km or more to reduce measurement noise. WIVERN will make a bridge between the synoptic scale and the mesoscale thanks to the wide swath and the high spatial resolution. In addition, WIVERN can play a synergistic role with Aeolus as the latter samples the

winds in clear sky, while WIVERN will sample in-cloud winds.

This is the first paper dedicated to the specific task of evaluating ~~The focus of this paper is to study, for the first time, the impact of assimilating WIVERN in-cloud winds in the Weather Research and Forecasting (WRF) Limited Area Model (LAM) model~~ (Skamarock et al., 2019). The impact of WIVERN in-cloud winds data assimilation in the global model ~~Action de Recherche Petite Echelle Grande Echelle (ARPEGE)~~ has been studied in (Sasso et al., 2025). ~~To robustly assess the impact of WIVERN wind data assimilation (DA) on WRF forecasts, we use an ensemble data assimilation (EDA) framework (Tan et al., 2007; Tan et al., 2013).~~

To assess robustly the impact of WIVERN wind DA on WRF forecasts, we use an En3DVar approach. In this approach the model error is computed (or partially computed depending if a hybrid approach is used) from the ensemble members and the model error is aware of the meteorological conditions of the day. Hamill and Snyder (2000) were the first to propose a 70 3DVar-based hybrid scheme in which the static background error covariance (BEC) in a 3DVar system was replaced by a linear combination of the static and ensemble-derived BEC. They also found that the analysis performs the best when BEC is estimated almost fully from the ensemble when the ensemble size was large (100 in their case). When the ensemble is smaller, the system benefits from a lesser weighting given to the ensemble-based covariances. Wang et al. (2009) also found that a hybrid 75 system based on an ensemble transform Kalman filter (ETKF) is more robust than EnKF for a two-layer primitive equation model when the ensemble size is small and when the model error is large. The good performance of En3DVar data assimilation systems was studied and reported in several other studies (Zhang et al. (2013); Lorenc et al. (2015); Li et al. (2012); Pan et al. (2014); Wang et al. (2009), for example).

In contrast, in the classical 3DVar approach, a static climatological background error covariance matrix is used, usually computed for a period which is representative of the period of interest. In a sensitivity test considered in this paper, the results of the 80 En3DVar approach are compared with the classical 3DVar to show the impact of using a background error matrix representative of error of the day compared to a background error matrix computed from the climatology.

To study the impact of WIVERN in-cloud winds data assimilation in the WRF model we selected a storm in the Mediterranean, that has long been described as a global cyclogenesis hotspot and one of the most sensitive regions in terms of global warming (Giorgi and Lionello, 2008). Mediterranean cyclones are key features of the region's climate and water cycle (Flaounas et al. 85 (2022); Ali et al. (2023)) and they have an important societal and economic impact. These cyclones are driven by the large-scale extra-tropical circulation and Rossby wave breaking, and they involve contributions from both dry dynamic and moist diabatic forcings (Flaounas et al., 2021; Raveh-Rubin and Wernli, 2016; D'Adderio et al., 2022; Lfarh et al., 2023). Among Mediterranean cyclones, a special class, known as Medicane (Mediterranean Hurricanes, Emanuel (2005)), has attracted considerable attention of both the scientific community and the public. For the broader public, these storms are important due to 90 their high destructive power. From a scientific point of view, they are of interest because they show tropical characteristics such as a symmetric structure, a cloud-free centre resembling an eye, and a warm core extending at least partially through the troposphere (D'Adderio et al., 2022; Di Francesca et al., 2025). There are many studies on Medicane (Fita and Flaounas (2018); Miglietta and Rotunno (2019); Dafis et al. (2020); ~~just to cite a few~~), that examine the physical characteristics of these storms and the key role of deep convection in the formation of the deep warm core. Among this type of storm, we selected

Medicane Ianos as a case study for two main reasons: a) it ~~This Medicane~~, was among the most intense Medicanes ~~in the Mediterranean~~ (Lagouvardos et al., 2022; D'Adderio et al., 2022) ~~occurred in mid-September 2020 and impacted Greece with strong winds, heavy precipitation, and storm surges during landfall (Ferrarin et al., 2023; Ferrarin et al., 2023)~~; b) Ianos it was used in several studies to investigate investigating various aspects of the storm, including ~~the influence of convective parameterisation and microphysical characteristics (Saraceni et al., 2023; Saraceni et al., 2021)~~, the impact of the sea surface 100 temperature on storm evolution (Varlas et al., 2023) ~~as well as~~ and diabatic forcing and storm surge impact (Sanchez et al., 2023; Ferrarin et al., 2023; Androulidakis et al., 2023). Moreover, the performance of different NWP model settings, including ~~the influence of convective parameterization and microphysical characteristics were studied for this Medicane (Saraceni et al., 2023; Comellas Prat et al., 2021)~~.

Recently, Pantillon et al. (2024) published a model inter-comparison of Medicane Ianos simulations, as part of the COST Initiative 105 CA19109 “MedCyclones: European Network for Mediterranean Cyclones in weather and climate”. They compared 10 different modeling systems, including different WRF model configurations, for storm track prediction. The study highlighted the importance of explicitly resolving convection in the simulation of the storm, which plays a fundamental role for accurately simulating cyclone track and storm deepening. ~~The authors~~ ~~It~~ also found a spread among the ensemble members, with most models predicting a storm track shifted southward compared to the best ~~a-posteriori~~ estimate (Flaounas et al., 2023). ~~WRF was 110 among these models.~~ In this paper, as in Pantillon et al. (2024), we use storm track as a primary metric to study the impact of WIVERN winds DA in the WRF model.

~~As stated, this paper studies the impact of WIVERN winds DA on the forecast of the Medicane Ianos.~~ Despite the study uses the WRF model as meteorological driver, it can be useful to a much wider community of NWP users as WIVERN observations are completely new and have a great potential to impact the quality of the NWP forecast; indeed not only we lack winds 115 observations compared to other data (Horányi et al., 2015; Sasso et al., 2025) and WIVERN will help to solve this issue, but WIVERN will take its observations within clouds, i.e. within features whose prediction plays a major role in weather forecast. The paper is organized as follows: Section 2 presents the WRF model configuration, the ensemble framework, and the methodology used to generate pseudo-observations for the case study. Section 3 shows the results of WIVERN winds data assimilation 120 on prediction of the Medicane Ianos trajectory and other storm parameters, as well as its impact on rainfall and surface wind prediction. This section also includes two sensitivity tests on the choice of background and observation error matrices. ~~In Section 3 we also investigate the potential of WIVERN winds DA for a forecast initialized when Ianos, yet in a mature phase, was far from the landfall.~~ Conclusions are provided in Section 4. Appendix A gives further details on the assimilation of WIVERN winds ~~in the En3Dvar approach used in this paper. using three-dimensional variational data assimilation (3DVar).~~

## 2.1 WRF model settings

In this work, we use the WRF model V4.1 with 400 grid points in both the west-east (WE) and south-north (SN) directions, and 55 vertical levels extending from the surface to 50 hPa (Skamarock et al. (2019)). The model horizontal grid resolution is 4 km ~~in both WE and SN directions~~, and the domain covers the Central Mediterranean basin. The center of the domain

is located at (15°E, 40°N) ~~, with the SW and NE corners at (6°E, 35°N) and (23°E, 46.6°N), respectively~~ (see Figure 1).

The physical parameterisation used in the model include the Thompson microphysics scheme (Thompson et al. (2008)), the Mellor-Yamada-Janjic turbulent kinetic energy boundary layer scheme (Janjic (1994)), the Dudhia scheme for shortwave radiative transfer (Dudhia (1989)), and the Rapid Radiative Transfer Model (RRTM) for longwave radiation Mlawer et al. (1997)).

Convection is assumed explicitly resolved and no cumulus parameterization is activated. The setting of the WRF model outlined above is the result of a compromise between the quality of the results and the computational resources. Comellas Prat et al. (2021), in a comparative study using different settings of the WRF model for the simulation of the Medicane Ianos, showed that a setting similar to that used in this paper gave the best result. In addition, the results of this model setting is well in line with the results of Pantillon et al. (2024), who used several NWP models, including WRF, to simulate the evolution of Ianos.

The different members of the WRF ensemble are generated taking initial and boundary conditions (IC/BC) ~~are taken~~ from the European Centre for Medium-range Weather Forecast - Ensemble Prediction System (ECMWF-EPS) of the Integrated Forecasting System (IFS) run issued at 12:00 UTC on 15 or 16 September 2020, depending on the numerical experiment considered. The ECMWF-EPS consists of one unperturbed (control) member and 50 perturbed members, resulting in a total of 51 WRF runs nested within the ECMWF-EPS initial and boundary conditions. As a result of this setting the WRF ensemble has 145 51 members and each member of the WRF ensemble is numerated following the ECMWF-EPS member that initialized WRF. The spatial horizontal grid resolution of the ECMWF-EPS for the Medicane Ianos is about 36 km.

## 2.2 Methodology

In the following we refer to the numerical experiment starting at 12 UTC on 16 September 2020, initialized from ECMWF-EPS 150 analysis/forecast cycle issued at the same time (see Figure 2 for a schematic of the simulations discussed in this paper). Other experiments, starting at different times are discussed in Section 3.4.

Two different data assimilation experiments ~~eyeles~~are considered: a 3-hourly cycle and a 24-hourly cycle. In the 3-hourly cycle, referred to as  $WIV_{3h}$ , WIVERN winds along the LoS are assimilated every 3h ~~for 48h~~. Although WIVERN would have sampled Ianos less frequently, ~~have a longer repetition cycle~~, this 3h experiment is used to assess the effectiveness of WIVERN 155 wind data assimilation under idealized conditions, assuming the satellite operates in a constellation formation. In addition, the  $WIV_{3h}$  experiment will be used to show some characteristics of the data assimilation. Hereafter, "WIVERN winds DA" refers to the assimilation of WIVERN in-cloud winds measured along the line of sight.

**Figure 1.** a) WRF domain; b) WIVERN track for the Medicane Ianos. The a-posteriori best estimated trajectory is shown by the red-line, the representative member trajectory is shown by symbols. The time range of the trajectories is indicated in the legend. The surface pressure along the track is shown by the color (both line and marks). The WIVERN sub-satellite point is represented by the blue dashed line, while the radar conical scan is shown by the gray dashed line, showing the 800 km wide swath. Times are indicated in local time (UTC+2h) ascending/descending node. The passages of WIVERN over the area would have occurred at 00 UTC on 18 and 19 September 2020 and at 12 UTC on 17 September 2020, however, only the 12 UTC observations would have sampled Ianos at about the center of the scene.

**Figure 2.** Schematic of simulations considered in this paper. The black lines are the CTRL forecast starting at 12 UTC on 15 September and 16 September. Green lines correspond to experiments with WIVERN winds DA. Each dot shows the assimilation time. WIVERN pseudo-observations are produced in correspondence of the assimilation time. No other observations are assimilated.

In the 24-hourly cycle, referred to as  $WIV_{24h}$ , a single assimilation of WIVERN winds is performed at 12 UTC on 17 September, when Ianos was already fully developed. This setup is designed to represent a more realistic scenario, in which WIVERN 160 overpasses a mature storm system. Finally, a control ensemble,  $CTRL$ , starting at 12 UTC on 16 September 2020 and running

for two days, is included as reference. This ensemble is run without any data assimilation (no other data are assimilated in the experiments considered in this paper, except WIVERN winds), using only different initial and boundary conditions derived from the ECMWF-EPS. ~~Each WRF ensemble member corresponds directly to the ECMWF-EPS member that provides its IC/BC.~~

165 For the data assimilation, we use the 3DVar developed by Federico (2013), based on the work of Barker et al. (2004) (see also Torcasio et al. (2024) for recent developments of the 3DVar software). For both  $WIV_{3h}$  and  $WIV_{24h}$  the background error covariance matrix is computed from the *CTRL* ensemble members at 12 UTC on 17 September 2020 by:

$$\mathbf{B} = \mathbf{XX}^T \quad (1)$$

$$\mathbf{X} = \frac{1}{(N_{ens} - 1)^{1/2}} (\mathbf{x}_{b1} - \overline{\mathbf{x}^b}, \mathbf{x}_{b2} - \overline{\mathbf{x}^b}, \dots, \mathbf{x}_{b_{N_{ens}}} - \overline{\mathbf{x}^b}) \quad (2)$$

where  $N_{ens}$  is the number of ensemble members and  $\overline{\mathbf{x}^b}$  is the ensemble average. Further details about the background error matrix and its implementation in the 3DVar are given in Appendix A. For a complete reference, the reader is referred to Federico (2013) and Barker et al. (2004). The date of the 12 UTC on 17 September was used to compute the background covariance error matrix for two main reasons: a) the WRF forecast is enough far from the initialization time and the model has 175 developed its own dynamic and thermodynamics characteristics; b) the spread of the ensemble is similar to that of Pantillon et al. (2024) at the same time, and the results of this paper can be representative of more general conditions, using an ensemble of different models.

To evaluate the impact of WIVERN Doppler data assimilation on the WRF forecast, we adopted the following steps:

- First, we generated an ensemble of the WRF model nested in the ECMWF-EPS of the IFS, starting at 12 UTC on the 16 September 2020 (*CTRL* ensemble);
- From the *CTRL* ensemble, we selected a representative (best) member, which is the one whose simulated storm track is in closest agreement with the best a-posteriori estimate of Ianos' trajectory provided by Flaounas et al. (2023). We used the storm track only to define the representative member as the pressure of the best estimated trajectory remains too high ( $> 1000$  hPa), compared to the observations from the Palliki meteorological station on Kefalonia island, which recorded a minimum pressure of 984 hPa (at 05 UTC on 18 September) ;
- We then generated pseudo-observations of WIVERN winds using the simulator forward operator developed by Da Silva et al. (2025) (see also Battaglia et al. (2022)). The simulator was applied to the output of the representative member. One and WIVERN pseudo-observations are generated every 3h, from 15 UTC on 16 September to 09 UTC on 18 September, from this member. These pseudo-observations consist of WIVERN winds along the LoS.

**Figure 3.** Trajectories followed by the ensemble members of the WRF *CTRL* ensemble. The black trajectory is the member 42.

### 190 2.3 Representative member choice

Figure 3 shows the 51 trajectories of the *CTRL* ensemble ~~Medicane Ianos, simulated by the WRF runs nested within the 51 members of the ECMWF-EPS~~. Each trajectory is defined by tracking the position of minimum sea-level pressure around the area of the Medicane. ~~The model output is saved every 1h and two consecutive sea-level pressure minima are connected by a segment~~. The color bar corresponds to the colors of the segments ~~joining two dots~~ and indicates the pressure at the initial point of each segment (~~there are three segments between two dots~~). The dots, plotted every 3h ~~for clarity~~, correspond to the position of the cyclone at different times (~~there are three segments between two dots~~). As expected, the trajectories are initially close to each other, but diverge with time, due to the amplification of small differences in the initial conditions by the evolving atmospheric flow. ~~This is confirmed by the spread of the ensemble, which steadily increases from 23.2 km at 00 UTC on 17 September to 60.4 km at 06 UTC on 18 September. The red line in Figure 3 shows the best a-posteriori estimated trajectory.~~

195 According to ~~this trajectory the best estimate~~, Medicane Ianos made landfall between the islands of Zakynthos and Kefalonia (Lagouvardos et al., 2022; Flaounas et al., 2023). ~~However, the pressure of the best estimated trajectory remains too high (>1000 hPa), compared to the observations from the Palliki meteorological station on Kefalonia island, which recorded a minimum pressure of 984 hPa (at 05 UTC on 18 September)~~. As stated in Section 2.2 the minimum surface pressure of the best estimated trajectory remains too high compared to observations. This discrepancy is because the reference trajectory is

200 derived from the ERA5 reanalyses, ~~whose horizontal resolution (30 km) smears out the minimum pressure at the center of the storm. This is confirmed by the fact that the ECMWF-IFS operational analysis, having a higher horizontal resolution (18 km), is about 10 hPa lower than the ERA5 reanalysis~~. Therefore, we only considered the trajectory, i.e. the position of the surface minimum pressure, and not the sea level pressure values, when comparing the WRF ensemble trajectories of the Medicane Ianos with the reference trajectory.

As shown in Figure 3, most simulated trajectories are displaced to the south of the best estimated trajectory. This result is in agreement with that of Pantillon et al. (2024), showing that the forecast of the Ianos trajectory of several meteorological models, including WRF, is to the south of the best estimated trajectory.

The representative member of the WRF ensemble was selected by minimizing the average distance between the members and the best estimated *a-posteriori* trajectory. This average distance is calculated as:

$$d_{be,m}(t) = |\mathbf{r}_{be}(t) - \mathbf{r}_m(t)| \quad (3)$$

$$\bar{D} = \sum_{t=1}^T \sum_{m=1}^M \frac{d_{be,m}(t)}{TM} \quad (4)$$

where  $\mathbf{r}_m$  and  $\mathbf{r}_{be}$  are the position vectors of the minimum sea-level pressure of the  $m$ -th ensemble member and the best estimated trajectory, respectively, at time  $t$ . The distance of the whole ensemble from the best estimated trajectory ( $\bar{D}$ ) is given 220 by averaging  $d_{be,m}$  over all ensemble members  $M$  and forecast times  $T$ . The trajectory error was computed over the period from 15 UTC on 16 September 2020 to 06 UTC on 18 September 2020, based on the hourly WRF output. This time range was chosen considering two requirements: a) the simulation starts at 12 UTC on 16 September, and the minimum pressure 225 at this initial time is determined by the ECMWF-EPS initialisation rather than by the WRF evolution; b) after 06 UTC on 18 September, many ensemble members made their landfall. To simplify the analysis, we consider the trajectory of the Ianos evolution over the Ionian Sea.

Figure ??, panel a), shows a histogram of The distances between each member of the *CTRL* ensemble and best estimated trajectory from Flaounas et al. (2023) were computed. The smallest distance, 30.0 km, is achieved by member 42, while the largest, 101.2 km, is achieved by member 11. Figure ??, panel b), shows The trajectory of member 42 is shown in black in Figure 3 together with the best estimated trajectory (i.e. the same trajectories of Figure 1). It is apparent that this member 230 follows well the best estimated trajectory of Ianos. According to the result of Figure 3, member 42 is chosen as the representative member and is used to generate pseudo-observations of the Medicane Ianos, as if it would have been observed by WIVERN. We computed pseudo-observations every 3 h, for a total of 15 WIVERN scenes, from 15 UTC on 16 September to 09 UTC on 18 September. This was done by applying the WIVERN simulator (Da Silva et al. (2025); Battaglia et al. (2022)) to the member 42 output files, and assuming that the Ianos Medicane is well-sampled by WIVERN. From now on, the member 235 42 represents our truth, i.e. the simulation to reproduce after applying WIVERN winds DA to other ensemble members.

## 2.4 WIVERN pseudo-observations

WIVERN has a conically scanning geometry with the antenna pointing at 41° off nadir. The antenna does 12 rpm and the winds along the LoS are a combination of the zonal, meridional and vertical wind components. The winds along the LoS are 240 given by:

**Figure 4.** WIVERN pseudo-observations at 2 (panel a), 5 (panel b) and 7 km (panel c). Each dot represents a pseudo-observation at the specific azimuth. The position of the minimum sea level pressure is approximately indicated by a black cross. The number of pseudo-observations are 1078 at 2 km, 1821 at 5 km and 2797 at 7 km

$$f(U, V, W) = U \sin \theta \cos \phi + V \sin \theta \sin \phi + W \cos \theta \quad (5)$$

where  $\theta$  is the angle between the WIVERN antenna and the vertical direction ( $41^\circ$ ),  $\phi$  is the azimuth, and  $U, V, W$  are the zonal, meridional and vertical wind components. Assimilation is done in stratiform areas where  $W$  is negligible. Stratiform areas are simply identified by the presence of a radar echo above a minimum threshold ( $-15\text{dBz}$ ), as the attenuation of the

radar signal in deep convective areas prevents the return of the echo from these areas (see below). In addition, for the setting of the WIVERN simulator used in this paper, we are considering observations averaged over a volumes of  $5$  by  $5\text{ km}^2$  in the horizontal and  $500\text{ m}$  thickness in the vertical direction and it is not common to have  $W$  velocities comparable to  $U$  and  $V$  velocities on these large volumes. Finally, there could be occurrences where/when we assimilated the WIVERN winds in volumes where  $W$  is not negligible compared to  $U$  and  $V$ , nevertheless we assumed  $W$  negligible. For stratiform areas the  
WIVERN winds along the LoS are given by:

$$f(U, V) = U \sin \theta \cos \phi + V \sin \theta \sin \phi \quad (6)$$

When generating WIVERN pseudo-observations, we assume that the Medicane is well sampled by WIVERN. Figure 4 shows the best a-posteriori estimated trajectory of Ianos, the representative member trajectory and the conical scan of WIVERN at 12 UTC on 17 September 2020. The assumed satellite ground track is shown in Figure 1 for all scenes used for generating  
pseudo-observations. With this assumption geometry, the Ianos trajectory center is well inside the radar swath, and the Medicane is well sampled by WIVERN.

Pseudo-observations generated applying the WIVERN simulator to the representative member 42 scene at 12 UTC on 17 September 2020 are shown in Figure 4 for three different vertical levels. Figure 4 displays also the position of the Ianos minimum sea level pressure, which is well inside the WIVERN swath, showing that the storm is well sampled. The value of the  
pseudo-observation is the component of the wind in the direction where the antenna is pointing. As shown by Eqn. (6), the

**Figure 5.** a) Vertical distribution of the number of WIVERN observations; b) Vertical average of the observations' error (black curve), of the model error (blue curve) and of the radiosoundings error used by the WRF data assimilation software (red curve). The model wind speed error is also used to inflate the observation error in a sensitivity experiment.

wind component along the LoS are dependent on the azimuth and on the angle between the radar antenna and the vertical direction ( $41^\circ$ ) and they do not have a straightforward physical interpretation.

As shown by Figure 4, Ianos is well sampled by WIVERN at 12 UTC on 17 September. However,  $WIV_{3h}$  simulation uses WIVERN pseudo-observations from 15 UTC on 16 September to 09 UTC on 18 September every 3h. As shown by Figure 1 265 panel b) the center of the Ianos Medicane, which shows a little tilting in the vertical as tropical cyclones, is well inside the WIVERN swath for all times.

Figure 5 panel b) shows the observation error (black curve), the model errors (blue curve) and the radio-sounding wind errors used in the WRF model (red line) as functions of altitude. The observation error  $\sigma_{LoS}^2$ , decreases with increasing signal

to noise ratio, as discussed in Battaglia et al. (2025b). For WIVERN winds along the LoS, the observation error is given by:

$$\sigma_{LOS}^2 = \frac{1}{N} \frac{v_{Nyq}^2}{2(\pi\beta)^2} \left[ \left( 1 + \frac{1}{SNR} \right)^2 - \beta^2 \right] \quad (7)$$

$$SNR = 10^{\frac{Z - Z_{min}}{10}}; \quad \beta = e^{-\frac{1}{2} \frac{\pi^2 \sigma_v^2}{v_{Nyq}^2}}; \quad v_{Nyq} = \frac{\lambda}{4T_{HV}}$$

where  $\lambda = 3$  mm is the radar wavelength,  $T_{HV} = 20\mu s$  is the separation between the two polarized pulses H and V,  $N = 40$  is the number of pulse pairs,  $\sigma_v = 4$  ms $^{-1}$  is the Doppler spectral width,  $v_{Nyq}$  is the Nyquist velocity, and  $Z_{min} = -15$  dBZ.

With these settings, WIVERN pseudo-observations are generated at 5 km horizontal resolution. The  $\sigma_{LOS}^2$  does not take into account errors like non uniform beam filling, wind shear, and mispointing errors (Scarsi et al. (2024b)). These errors are expected to be lower than 1 ms $^{-1}$  (Battaglia et al. (2022); Tridon et al. (2023); Battaglia et al. (2025a)). However, to account for these errors the corrected observation error  $\sigma_{cLOS}^2$  is:

$$\sigma_{cLOS}^2 = \sigma_{LOS}^2 + c_1^2 \quad (8)$$

with  $c_1 = 1$  ms $^{-1}$ . In summary, the error associated to each pseudo-observation is given by Eqn. (8) and is a function of the the signal-to-noise ratio, which is a function of the reflectivity. The observation error covariance matrix is assumed diagonal and pseudo-observations are thinned to 10 km to account, at least partially, for assuming a diagonal observation matrix. and the error variancee associated with each observation is given by Eqn. 8.

In WIVERN simulator the winds along the LoS are generated starting from the WRF wind components of the member 42 and applying Eqn. (6). The noise is added to the observation through Eqn. (8), which depends on the signal-to-noise ratio. To have wind pseudo-observations along the LoS, a minimum reflectivity of -15 dBZ must be observed. The reflectivity is computed from the distribution of the WRF hydrometeors simulated by member 42. It follows that pseudo-observations are not generated if clouds are optically too thin or, as the simulator accounts for the radar signal attenuation, if the radar beam is attenuated, as in deep convective areas. Figure 5 panel a), shows the number of pseudo-observations available at different vertical levels 290 at 12 UTC on 17 September, which is particularly important as it is the time when WIVERN winds are assimilated in WRF for  $WIV_{24h}$  experiment. The largest number of observation is at 7 km height, which decreases at higher levels due to the reduced optical thickness of clouds, and at lower levels due to radar signal attenuation.

Figure 5 panel b) shows the wind observation error averaged for each vertical level (black curve), the model wind speed error (blue curve) and the radio-sounding wind errors used in the WRF data assimilation (red line) as functions of altitude.

The comparison between the WIVERN LoS error and the radio-sounding error shows the good performance of WIVERN observations, whose error is lower than radio-sounding above 4.5 km.

Figure 5 panel b) shows that The model wind speed error is given by the square root of squared error for the zonal and meridional wind components. These errors are given, for each level, by the diagonal elements of the vertical component of the background error matrix ( $\mathbf{B}_z$ , see Appendix B of Federico (2013) for details). The model wind speed error is larger than the 300 observation error; this suggests a large impact of WIVERN data assimilation for the Ianos case study. To study the sensitivity of the results to the assumed observation error, Section 3 presents an experiment with in which the observation error is inflated to

match the model error. ~~As for the En3DVar framework, the observation error for each measurement is computed following the Eqn. (8). The observation error matrix is assumed diagonal, i.e. we neglect the error correlation among different observations. WIVERN pseudo-observations are generated with the WIVERN simulator with an horizontal resolution of 5 km, and they.~~

Finally, no data thinning is applied in the vertical direction and pseudo-observations are assimilated starting from 1 km above the sea (the largest part of the pseudo-observations assimilated) and 2 km above the land, to account for the ground clutter.

### 3 Results

#### 3.1 Impact on trajectory, surface pressure and winds forecast

After generating pseudo-observations, we performed a run of the whole ensemble assimilating WIVERN pseudo-observations every 3 h. Before showing the results of assimilating WIVERN winds on the WRF forecast, it is important to assess how 310 WIVERN observations are being assimilated. For this purpose, we consider observation diagnostics in the form of mean innovation (e.g., bias), root-mean-square innovation (RMSI) and the total number of data assimilated for each assimilation cycle (Jones et al., 2016). Innovation and RMSI are calculated by taking the difference between prior and posterior fields  $H(\mathbf{x})$  and comparing against observations  $\mathbf{y}$ :

$$INNOV = \mathbf{y}_n - H(\mathbf{x}_n) \quad (9)$$

and:

$$RMSI = \sqrt{\left( \sum_{i=1}^N INNOV_i \right) / N} \quad (10)$$

where N represents the number of pseudo-observations assimilated.

These statistics are reported in Figure 6. The RMSI slowly increases from the start of the simulation until the end. This 320 reflects the increase of the spread of the ensemble. Close to the end of the forecast, there is an increase of the RMSI as a consequence of the increased spread of the trajectories approaching the landfall. The RMSI is reduced by 15% through WIVERN DA at the start of the forecast; this percentage decreases as the forecast progresses (down to 5%) then it increases again close to the end of the forecast, when Ianos approaches the landfall (15-20%). The bias remains small before and after DA, showing that WIVERN pseudo-observations are unbiased. Moreover, WIVERN DA tend to reduce this small bias. The 325 number of pseudo-observations assimilated are about 50% of the total number of pseudo-observations available; this is expected because a data thinning of 10 km is applied at each level and pseudo-observations are generated at 5 km horizontal resolution. The result of assimilating WIVERN winds every 3h is shown in Figure 7. The assimilation of WIVERN winds every 3 h ~~has a very important impacts on~~ the Ianos trajectory ~~in two ways~~: first, all the trajectories are now focused along the trajectory of the representative member 42, which is shown in black and without dots for clarity; second, the landfall occurs in the northern part

**Figure 6.** Innovation (m/s), RMSI (m/s), number of pseudo-observations (NDATA) available for each assimilation cycle and number of pseudo-observations assimilated (NDATA\_ASSIM) for each assimilation cycle for the  $WIV_{3h}$  experiment. The left y-axis refers to the innovation and RMSI while the right y-axis refers to the number of pseudo-observations and to the number of pseudo-observations assimilated.

of the Peloponnese with many member crossing the Zakynthos island or passing through the gap between the Kefalonia and Zakynthos islands. The  $\bar{D}$  distance of the ensemble from the member 42 computed by Eqn. (3-4) is 14.9 km, to be compared with 62.5 km of the *CTRL* ensemble (Table 1).

Importantly, the En3DVar used in this paper neglects cross correlation among variables and only zonal and meridional wind components are adjusted by the DA. Nevertheless, the other atmospheric variables are adjusted by the model physics. For 335 example, the cyclostrophic equilibrium is important for Medicanes and, once velocities are adjusted by DA, the pressure field is adapted to the adjusted winds. These changes propagates to other variables through the model physics and, in general, the model follows a different trajectory in the phase space after DA. changes in the winds caused by the WIVERN winds DA affect are propagated, through the model physics, to other parameters, focusing the ensemble members towards the representative member 42. This point is demonstrated, for example, for the minimum sea level pressure in Figure 8, with the spread of the 340 minimum sea level pressure substantially reduced by the WIVERN winds DA. It is also noted the increase of the minimum sea level pressure of the  $WIV_{3h}$  ensemble forecast (Figure 8 panel b) compared to the *CTRL* ensemble forecast (Figure 8 panel a) as many members of the *CTRL* ensemble were predicting a pressure lower than the member 42. and this is improved by WIVERN winds DA.

It is important to note that the current setup of  $WIV_{3h}$  has a very short spin-up time (3h). Since the WRF model is run on a 345 single, relatively small domain at convection-permitting resolution, the atmospheric fields initialized from the coarser ensemble forecast could produce imbalances in the WRF forecast, lowering its quality. While a longer spin-up time will be adopted in future studies, we did a sensitivity experiment to evaluate the impact of the short spin-up time for the  $WIV_{3h}$  experiment. We

**Figure 7.** Trajectory followed by the WRF ensemble when WIVERN DA is applied every 3 h. The member in black is the representative member 42.

chose randomly 20 members of the ensemble and we assimilated the WIVERN winds starting from 00 UTC on 17 September allowing for a 12h spin-up time. Results for the 20 members (not shown) are similar to those of Figure 7, in the sense that the 350 trajectories of the ensemble strictly follow the trajectory of member 42 after 12h from the first DA time, i.e. 12 UTC on 17 September. So, for the specific case study and settings of this paper, the short spin-up time used by  $WIV_{3h}$  does not impact the quality of the results.

Hereafter we focus on the  $WIV_{24h}$  numerical experiment and two sensitivity tests. In this experiment, data assimilation is done at 12 UTC on 17 September for each member of the ensemble, then follows a 24 h forecast. The motivations for choosing the 12 UTC on 17 September are: a) the storm is well formed, so we can generate a good pseudo-scene; b) the landfall is enough far (18 h before the landfall) and the forecast is of practical importance.  $WIV_{24h}$  represents a realistic scenario in which WIVERN sample Ianos one-time in the period of the simulations considered in this paper. Indeed, considering the 1.5 day revisiting time at the equator of WIVERN and the fact that Ianos lasted few days, the Ianos Medicane would have been sampled at least once. We chose the 12 UTC on 17 September for the reasons stated above; in this scenario, the Ianos Medicane 360 would have been sampled two times (Figure 1 panel b) in the time window considered in this work, but the second time would have been too close to the landfall for the forecasting purposes of this paper.

The trajectories of  $WIV_{24h}$  ensemble are shown in Figure 9. By comparing Figure 9 and Figure 3, it is apparent the positive impact of WIVERN DA on the forecast of the Ianos trajectoryies. In particular, the *CTRL* forecast, Figure 3, shows several trajectories going towards the southern part of the Peloponnese; these trajectories are shifted northward in the experiments with 365 WIVERN DA, even if the  $WIV_{24h}$  trajectories still tend to go to the south of the member 42 trajectory. Another interesting point is the increase in pressure along the final part of the trajectories compared to the *CTRL* ensemble. This is shown by the yellow-green colors of the segments in the last part of the trajectories in Figure 9, compared to the green colors of the same traits in Figure 3. This difference is attributed to the change in the storm dynamics after DA, and to the WRF model physics,

**Figure 8.** Time variation of minimum sea level pressure for a) the background ensemble; b) the  $WIV_{3h}$  experiment. The black line in panel a) is the minimum sea level pressure of member 42.

that propagates changes in the wind field to the mass fields through physical relationships.

In Figure 9, the violet arrow indicates the time of analysis. It is interesting to note that the trajectories converge after the analysis, as the different ensemble members tend towards the representative member 42. These results are summarized in Table 1, for the period 12 UTC on 17 September to 06 UTC on 18 September, showing the  $\bar{D}$  of the ensemble from the member 42 for the  $CTRL$ ,  $WIV_{3h}$  and  $WIV_{24h}$  ensembles. The table also includes the skill of the  $WIV_{3h}$  and  $WIV_{24h}$  ensembles compared to  $CTRL$ . The skill score,  $I(\%)$ , is given by:

$$I(\%) = 100 \frac{\bar{D}_{CTRL} - \bar{D}_{exp}}{\bar{D}_{CTRL}} \quad (11)$$

where  $\bar{D}$  is given by Equation (4) and  $exp$  can be 3h or 24h.

It is interesting to examine the trajectory error as a function of the ensemble member (i.e. Eqn. (4) without averaging over members) and as a function of time (i.e. Eqn. (4) without averaging over time). These statistics are shown in Figure 10a.

**Figure 9.** Trajectories followed by the members of the WRF ensemble  $WIV_{24h}$ . The trajectory in black is the representative member 42, while the violet arrow indicates the assimilation time.

Considering the error as a function of ensemble member, there are a few points to note. First, the  $WIV_{3h}$  ensemble performs  
 better than  $CTRL$  across all members. , the only exception being member 34, for which the improvement is very small. This shows that assimilating WIVERN every 3h gives a strong constraint on storm evolution for all the members. Assimilating WIVERN only once ( $WIV_{24h}$ ) still has a positive impact on the simulation of the Medicane Ianos' trajectory, with the error reduced for almost all members. Sometimes, the error is substantially reduced, as for the member 11 (and many others), sometimes the improvement of the trajectory forecast is small, as for member 47, and few times there is a negative impact of  
 assimilating WIVERN winds, as for member 35. However, the positive impact is very important for some members, with the trajectory error halved in some cases and improvements often larger than 15 – 20 km, and, when there is a negative impact of assimilating WIVERN winds, it is less than 10 km.

Member 34 deserves a special mention, as WIVERN DA does not improve its trajectory forecast. As shown in Figure 10 panel a), this member already has a very low background error (about 10 km), leaving little room for further improvement through 390 data assimilation. Overall, the analysis of Figure 10, panel a), leads to three main conclusions: a) assimilation of WIVERN winds along the LoS improves the forecast of the Ianos trajectory; b) the magnitude of the improvement varies depending on the ensemble member; and c) the negative impact of WIVERN DA is limited to few cases and less than 10 km, while the positive impact is often larger than 15 – 20 km. In addition, while the improvement of WIVERN DA is dependent on the member, this dependence is greatly reduced if WIVERN is assimilated every 3 h, showing the ability of WIVERN to substantially change  
 the storm evolution, when frequently assimilated.

Figure 10 panel b) shows the trajectory error as a function of forecasting time from 15 UTC on 16 September to 06 UTC on 18 September. The error of the  $CTRL$  ensemble increases with the forecast time, as expected. The  $WIV_{24h}$  forecast differs from the  $CTRL$  forecast after the analysis (12 UTC on 17 September). The improvement given by WIVERN winds DA is

**Table 1.** Average distances of the ensemble members from the member 42, and improvement (%) with respect to *CTRL* ensemble.

| <i>EXP</i>               | <i>Err(km)</i> | <i>I(%)</i> |
|--------------------------|----------------|-------------|
| <i>CTRL</i>              | 62.5           | /           |
| <i>WIV<sub>24h</sub></i> | 35.4           | 43          |
| <i>WIV<sub>3h</sub></i>  | 14.9           | <b>6476</b> |

large and lasts until the end of the period considered, showing a long-lasting effect of WIVERN DA. This point will be further investigated in Section 3.4.

**Figure 10.** Average error of the ensembles *CTRL*, *WIV<sub>24h</sub>*, and *WIV<sub>3h</sub>* as function of a) the member; and, b) the time. In the panel b), diamonds are shown every 3h while the square is every 24h from the simulation start.

To study the impact of WIVERN DA on the WRF forecast in more detail, we focus on the member 10, one of the members showing a substantial impact of WIVERN DA. Figure 11 shows the application of the 3DVar DA to this member the impact of

3DVar on both zonal and meridional wind components; indeed, considering that WIVERN pseudo-observations are a combination of zonal and meridional wind components, it seems interesting to show the impact of WIVERN winds DA on both wind components. The background (panel a) represents the strong winds associated with the Medicane Ianos, with meridional wind speeds **in the lower troposphere** reaching up to about  $40 \text{ ms}^{-1}$ , and a cyclonic circulation around the storm center (bipolar structure). After assimilation (panel b), the winds are still very intense and the cyclonic circulation well represented **as in the background**; however, the whole circulation has shifted several tens of kilometers to the east (refer to the longitude  $17.5^\circ E$ ).

The difference between the two fields has a “tripolar” pattern (panel c). From west to east, the meridional wind difference is positive to the west of  $17.5^\circ E$ , negative between  $17.5^\circ E$  and  $19^\circ E$ , and then positive again towards the east. This pattern corresponds to the net eastward shift of the storm center, and to a small reinforcement of the meridional wind component.

The vertical cross-section of the zonal wind difference (analysis minus background) is shown in Figure 11 panel d). It highlights a main dipolar pattern close to the storm center (around  $38.5^\circ N$ ), with negative values reaching up to  $20 \text{ ms}^{-1}$  at about 2500 m a.s.l. This dipolar pattern reaches **up to** 6 km height and is a consequence of the eastward shift of the storm center in the analysis. Interestingly, despite there is a decrease of the number of pseudo-observations in the lower troposphere (Figure 5a) caused by the attenuation of the radar signal, the number of pseudo-observations is enough to produce a substantial change of the circulation in the lower troposphere. In the upper troposphere, the difference between the analysis and background becomes more complex with several localized positive and negative values. This pattern is determined by observations at these levels and by the vertical structure of the background error matrix. Smaller differences are seen further north; they are caused by pseudo-observations over the NE of the domain (Figure 4).

The impact of WIVERN winds DA is long lasting; Figure 12 shows the 18 h forecast of the sea level pressure and surface winds starting from the analysis of Figure 11 is shown in Figure 12. The impact of WIVERN DA on the evolution of this member is **very** high. First, the position of the storm is much improved by WIVERN DA; while the member 42 (panel a) has just crossed the Zakynthos island, the storm center of member 10 without WIVERN DA is in the open sea (about 100 km to the SW of Zakynthos). The storm center of the member 10 after DA is close (about 10 km kilometers) to Zakynthos. Importantly, the minimum sea level pressure is also adjusted **with** by WIVERN DA; it is about 983 hPa for member 42 and for member 10 after DA, while it is lower than 975 hPa for member 10 without data assimilation. Similarly, the maximum wind speed at the surface of member 10 with WIVERN DA ( $28.1 \text{ ms}^{-1}$ ) is much closer to the representative member 42 ( $31 \text{ ms}^{-1}$ ) compared to the member 10 without WIVERN DA ( $39.8 \text{ ms}^{-1}$ ). In summary, Figure 12 shows that the assimilation of WIVERN winds changes the evolution of the storm not only for the trajectory but also for the physical characteristics, providing a **more realistic** representation of the Medicane Ianos closer to member 42. It is important to note that, as we assimilated WIVERN winds pseudo-observations derived from member 42, this member becomes our truth and being closer to it is equivalent to have an improvement of the forecast.

### 3.2 Impacts on the precipitation and surface winds forecasts

In this section we discuss the impact of assimilating the WIVERN winds along the LoS on the prediction of the precipitation and surface winds **in Kefalonia**. As stated before, member 42 represents our truth and comparison is done against its output,

**Figure 11.** Analysis of wind components at 12 UTC on 17 September 2020 for member 10: a) background meridional wind component at about 2500 m a.s.l; b) analysis of the meridional wind component at about 2500 m a.s.l; c) difference between analysis and background fields of the meridional wind component (same level of panels a-b); d) cross-section of the difference between the analysis and the background of the zonal wind component along the red line of panel c). The y-axis of panel d) shows the vertical levels and labels on the right y-axis correspond to the approximate height of the levels.

and no real observations are considered in this section. Figure 13 shows the precipitation accumulated from 12 UTC on 17 September to 12 UTC on 18 September for the *CTRL* ensemble (average), the representative member 42, and the *WIV<sub>24h</sub>* ensemble (average). The precipitation accumulated by the member 42 clearly mirrors the trajectory followed by the cyclone 440 with accumulated rainfall larger than 300 mm day<sup>-1</sup> in a swath oriented from SW to NE, ending over the Kefalonia island. The *CTRL* ensemble shows a precipitation pattern which is oriented in the NW-SE direction, differently from the pattern of the member 42. The field of *CTRL* is smoothed compared to the representative member, but this feature is caused by the average operator. The precipitation on the island of Kefalonia is largely underestimated compared to member 42 because the

**Figure 12.** Sea level pressure and surface winds (every 20 grid points) at 06 UTC on 18 September, i.e. 18 h after the assimilation time, for a) member 42 (from the CTRL forecast, i.e. without DA); b) member 10 (from the CTRL forecast, i.e. without DA); c) member 10 after the assimilation of WIVERN winds at 12 UTC on 17 September. The contour inside the Medicane eye in panel b) corresponds to the 975 hPa isobar.

rainfall is  $150 \text{ mm day}^{-1}$  for *CTRL* compared to more than  $300 \text{ mm day}^{-1}$  of the forecast for the member 42.

The rainfall accumulated by the ensemble *WIV*<sub>24h</sub> is in better agreement with the representative member 42 compared to *CTRL* because the intense precipitation swath is better oriented in the SW-NE direction and the rainfall predicted over Kefalonia is greater than  $300 \text{ mm day}^{-1}$ . This The better forecast of the *WIV*<sub>24h</sub> compared to *CTRL* is confirmed by the RMSE, calculated with respect to member 42 and averaged over the area defined by the longitudes  $17.5\text{--}22.5^\circ\text{E}$  and by the latitudes  $36.0\text{--}39.0^\circ\text{N}$  (red rectangle of Figure 13, panel c), which decreases from 51.0 mm of *CTRL* to 40.5 mm of *WIV*<sub>24h</sub>.

Figure ?? We also considered shows the errors of the ensembles *CTRL* and *WIV*<sub>24h</sub> calculated with respect to the member

**Figure 13.** Rainfall accumulated from the 12 UTC on 17 September to 12 UTC on 18 September by: a) the *CTRL* ensemble (average); b) the representative member 42; c) the *WIV*<sub>24h</sub> ensemble (average). The trajectory of the representative member 42 is shown in black.

42 for the surface winds for the coordinates corresponding to the island of Kefalonia ( $20.7^\circ\text{E}$ ;  $38.15^\circ\text{N}$ ) at 06 UTC on 18

September (not shown). The time is that of Figure 12 and the surface wind simulated by the member 42 in correspondence of the selected position is from NE with an intensity of  $26 \text{ ms}^{-1}$ . The members of both ensembles, *CTRL* and *WIV<sub>24h</sub>* are underestimating the wind intensity in Kefalonia. However, the underestimation is greater for the *CTRL* ensemble compared to *WIV<sub>24h</sub>*. ~~, as shown by the error distributions of Figure ?? panel a), which is more skewed towards negative values for the *CTRL* ensemble.~~ For example, 26 members of the *CTRL* ensemble underestimate the wind intensity by more than  $14 \text{ ms}^{-1}$ , while this number is reduced to 4 for the *WIV<sub>24h</sub>* ensemble.

~~The statistics for the whole ensemble are shown in Table ??.~~ Even if less apparent, similar considerations apply to the wind direction, ~~shown in Figure ?? panel b).~~ The *CTRL* ensemble has more members with errors larger than  $75^\circ$  compared to *WIV<sub>24h</sub>*. For a more complete analysis of the impact of WIVERN winds DA for the prediction of the surface winds in Kefalonia, the reader is referred to Federico et al. (2025). ~~The Bias of the direction is positive and winds are coming more from E-SE in the *CTRL* and *WIV<sub>24h</sub>* ensembles compared to member 42. This is coherent with simulating the storm center to the southwest of the member 42.~~

### 3.3 Sensitivity to model and observations error

In this section we consider the results of two sensitivity tests: in the first test we inflated the observation error, in the second test we changed the background error matrix. For the first experiment the observation error is assumed to be equal to the model wind speed error and dependent only on height. The observation error is shown in Figure 5 (panel b, curve *E2*) ~~and corresponds to the situation in which the analysis gives an equal weight to the background and to the observations. Considering the error of Figure 5 this sensitivity test~~ and roughly corresponds to inflating the WIVERN error by a factor of  $1.5 - 2$ ; we will refer to 470 this experiment as *E2*.

In the second sensitivity experiment, we changed the background error matrix, which was computed applying the NMC (Parrish and Derber, 1992) method to the period 1 September 2020 - 30 September 2020. Specifically, the background error matrix was computed from the difference of two forecasts verifying at the same time (both 12 UTC and 00 UTC) for the whole period. The background error matrix was computed using the operational analysis/forecast cycle issued by the ECMWF at 00 UTC and 12 475 UTC as initial and boundary conditions of the WRF model. In this configuration the background error matrix is representative of the meteorological conditions of the month of September 2020, while in the approach used for the *WIV<sub>24h</sub>* experiment, Eqn. (1), the background error matrix is representative of the error of the day. This sensitivity experiment will be referred to as *NMC*.

Results of both experiments are represented by the distribution of the trajectory errors of the ensemble members, i.e.  $\bar{D}$  without averaging over members, shown in Figure 14. The largest error correspond to the *CTRL* experiment, followed by *NMC*, *E2*, and *WIV<sub>24h</sub>* experiments.

The assimilation of WIVERN winds has, in general, a positive impact on the prediction of the Medicane Ianos trajectory as the experiment with doubled observation error has a performance much closer to *WIV<sub>24h</sub>* than to *CTRL*. Specifically, the average error of the experiment *E2* is ~~34.4 km~~, very similar to that of *WIV<sub>24h</sub>* ensemble (both about ~~34.4 km~~), however the 485 median error is larger (35.4 km) for *E2* compared to *WIV<sub>24h</sub>* (32.7 km). Similarly, the spread of the ensemble error is larger

for  $E2$  (14.5 km) compared to  $WIV_{24h}$  (13.3 km). Statistics for the trajectories error distributions are summarized in Table 2.

To better understand the low sensitivity of this case study to the observation error, Figure 15 shows the BIAS and RMSE for

**Figure 14.** Trajectories error distribution, respect to the representative member 42, of the ensembles  $CTRL$ ,  $NMC$ ,  $WIV_{24h}$  and  $E2$ . The boxes show the 25<sup>th</sup> and the 75<sup>th</sup> percentile, the black line inside the box is the median and the maximum and minimum values are the extremes of the error bar.

the first guess as a function of the vertical level. The first guess error (RMSE) is larger than the observation error and similar to the model wind speed error (Figure 5, panel b), and inflating the observation error has a relatively small impact on the 490 WIVERN winds DA performance. However, it is important to point out that winds around Medicane are very intense, and a wrong positioning of the storm results in high first-guess errors for the winds; in general it is expected that the role of the WIVERN observation error is larger than that reported for the Ianos case study.

Changing the background error matrix for this specific case has a notable impact on the trajectory forecast. This is expected in some measure, as the physical characteristics of the Medicane are rather different from those of the circulation of the period, 495 and choosing a climatological and static background error matrix is suboptimal compared to choosing a background error matrix aware of the specific error of the day. Specifically, the trajectory error averaged over the whole ensemble for the  $NMC$  experiment is 44.7 km (35.4 km for  $WIV_{24h}$ ), the median is 42.7 km (32.7 km for  $WIV_{24h}$ ) and the spread of the ensemble is 16.9 km (13.3 km for  $WIV_{24h}$ ). All these statistics show the sensitivity of the WIVERN DA impact to the choice of the background error matrix (Table 2).

### 3.4 Numerical experiments for different initialization times

In this section we consider the result of two experiments in which the ensemble is initialized one day before compared to that of previous sections, i.e. at 12 UTC on 15 September, and two analysis/forecasts are considered: a) assimilation at 12 UTC on 16 September and forecast from 12 UTC on 16 September to 00 UTC on 18 October (36 h forecast, Figure 2), experiment

**Figure 15.** Observation minus background errors [m/s] for the winds along the line of sight as a function of the vertical levels.

**Table 2.** Statistics of the trajectories error distribution, respect to the representative member 42, of the ensembles *CTRL*, *NMC*, *WIV<sub>24h</sub>* and *E2*.

| <i>EXP</i>               | Mean [km] | Median [km] | Spread [km] |
|--------------------------|-----------|-------------|-------------|
| <i>CTRL</i>              | 62.5      | 64.4        | 24.2        |
| <i>NMC</i>               | 44.7      | 42.7        | 16.9        |
| <i>WIV<sub>24h</sub></i> | 34.4      | 32.7        | 13.3        |
| <i>E2</i>                | 34.4      | 35.4        | 14.5        |

*WIV<sub>24h16</sub>*; and b) assimilation at 00 UTC on 16 September and forecast from 00 UTC on 16 September to 00 UTC on 18 October (48 h forecast, Figure 2), experiment *WIV<sub>12h</sub>*. Also, in the first experiment pseudo-observations are assimilated 24 h after the ensemble initialization, while in the second experiment WIVERN pseudo-observations are assimilated after 12 h from the ensemble initialization. The technique, the WRF and the data assimilation settings are those discussed in previous sections, with the following differences: a) when generating pseudo-observations, the track of the WIVERN satellite (Figure 1 panel b) 510 was shifted by 2 degrees to the west to center the storm position; b) the model output was saved every 3h and there are two traits between two dots; c) the background error matrix was computed at 12 UTC on 16 September 2020, i.e. 24 h after the ensemble initial time. We also consider the result of the *CTRL* forecast, starting at 12 UTC on 15 September 2020 (Figure 2). Figure 16 panel a) shows the result of the *CTRL* forecast. The best forecast, whose trajectory is in best agreement with the a-posteriori estimated trajectory of Ianos (Flaounas et al. (2023)), is that of the member 32 whose track is depicted in black. 515 Looking to the results of Figure 16 panel a) there are two main points to highlight: a) the spread of the trajectory is much larger than that of Figure 3; b) Many trajectories go south of Greece and towards the Aegean Sea and, for there trajectories,

the surface pressure remains greater than 1000 hPa. These two points show the lower predictability of the Medicane Ianos on 15 September compared to 16 September; more specifically, from Figure 16 panel a) it is not clear if the storm would have deepened, and the spread of the trajectories remains wide to take precise actions.

Both uncertainties are solved by the assimilation of WIVERN pseudo-observations in the  $WIV_{24h16}$  forecast. The trajectories of this ensemble are all going towards the western Peloponnese and the storm is deepening. The trajectories of the  $WIV_{12h}$  forecast are shown in Figure 16 panel c); all trajectories, with just one exception, approach the western and southern Peloponnese; again most of these trajectories show a clear deepening of the cyclone improving the forecast of the *CTRL* ensemble. All in all the results of this section show that WIVERN would have been able to constrain the forecast of the Medicane Ianos 525 also for the ensemble issued on 15 September 2020, giving a clear suggestion, with more than a day of advance, that the Medicane would have made its landfall in western/southern Peloponnese and that the storm would have deepened. It is also noticed that the forecast of the Medicane Ianos can be refined by the WIVERN wind observations DA as the Medicane is approaching the landfall; this is clearly shown by the comparison of the ensemble forecasts of Figure 3, Figure 9 and Figure 16 panels b) and c).

**Figure 16.** Trajectories of the Medicane Ianos for the experiment starting at 12 UTC on 15 September; a) *CTRL* ensemble; b)  $WIV_{24h16}$ ; c)  $WIV_{12h}$ . In b) the assimilation is done at 12 UTC on 16 September; in c) the assimilation is done at 00 UTC on 16 September. The trajectory in black is that of the reference member 32.

## 530 4 Conclusions

In this paper we considered the assimilation of WIVERN winds along the Line of Sight (LoS) in the WRF model for the case study of the Medicane Ianos. The assimilation is done using an En3DVar approach in an ensemble context and pseudo-observations are generated from one member of the ensemble whose trajectory was in best agreement with the best a-posteriori estimated trajectory of Ianos.

Two En3DVar assimilation cycles are considered: 3 h and 24 h. The first case, with very frequent data assimilation cycles, is used to verify the proper setting of the En3DVar and WRF model; it could represent a realistic condition if a constellation of many (4) four WIVERN satellites were operated. The second case corresponds to the realistic situation in which WIVERN

samples the Ianos storm once. The *WIV<sub>1</sub>2h* simulation also assimilated WIVERN winds once.

Results show an important impact of WIVERN wind DA on the WRF forecast. Considering the trajectory forecast, the average error is improved by more than 40% and the error decreases from 62.5 km to 35.4 km. It is shown that the trajectory forecast is improved for 46 out of 50 members ~~and that this improvement lasts at least 18 h~~. The forecast improvement is not confined to the trajectories but it is transferred from the dynamic to the mass field through the model physics, as shown by the improvement of the sea level pressure forecast and surface wind speed. This consideration applies also to the rainfall forecast which, with WIVERN DA, is more in agreement with that of the representative member for both pattern and intensity. ~~Similarly, the wind forecast for the Kefalonia island is improved for both speed and direction.~~

~~Finally, we~~ We presented the results of two sensitivity tests changing the observation and background errors. For the specific case of the Medicane Ianos, the impact of changing the background error matrix has a larger impact. This is caused by two main reasons: a) the first guess error is larger than the background and observation errors, also when the latter is inflated and the impact of assimilating WIVERN is expected high; b) ~~as~~ Medicanes are storms with peculiar characteristics ~~and~~ the background error matrix derived from the ensemble and representative of the “error of the day” is more appropriate for DA than the background error matrix computed for the whole period.

We considered the results of two additional forecasts starting at 00 UTC and 12 UTC on 16 September. For these forecasts, the background is given by the *CTRL* forecast starting at 12 UTC on 15 September, which shows a larger degree of uncertainty compared to the *CTRL* forecast issued on 16 September at 12 UTC. Indeed not only the spread of the trajectory is larger for the *CTRL* forecast issued at 12 UTC on 15 September, but there are a number of trajectories traveling south of Greece and going towards the Aegean Sea. These trajectories do not simulate the deepening of the storm. The impact of WIVERN winds DA is significant: not only the trajectories are predicted to go towards western Peloponnese, but the deepening of the storm is consistently predicted.

Recently, Pantillon et al. (2024) published a paper showing a model intercomparison for the Medicane Ianos forecast with 10 models participating to the comparison (including WRF). One of the aspects considered in the paper is the simulation of the Ianos trajectory from different models. Results show a spread of the models trajectories which is in line with the results shown by the *CTRL* ensemble; in addition most of the trajectories go to the south of the best *a-posteriori* estimated trajectory followed by Ianos, in agreement with the results of the *CTRL* ensemble. ~~The results of this paper show that the WIVERN winds DA has the potential to narrow the spread among different models in the forecast of such events. In this sense, this study is of interest not only for modelers using WRF, but also for the wider community of NWP users and developers because it shows that the assimilation of WIVERN winds, when available, has a potential to improve the forecast at long time ranges (24-48h). In this sense, the results of this paper show that the WIVERN winds DA has the potential to narrow down the spread among different models in the forecast of such events.~~

While the performance of WIVERN DA for the Ianos case study is promising ~~there are few points that limit the results of this work further research is needed~~. First, we considered only one case study and no general conclusions can be derived on the performance of the WIVERN DA at the regional scale. ~~This is valid not only considering other storm types, as extra-tropical cyclones in the Mediterranean, but also Medicanes, as they are highly variable in structure, environment, and predictability,~~

and it remains unclear whether the demonstrated benefits of WIVERN wind assimilation would hold across other cases with different dynamical regimes or forecast challenges. Second, we assimilated WIVERN data only, neglecting the global observing system, which is, however, considered indirectly from the ECMWF initialisation. This could result in an overestimation of the WIVERN impact on the forecast of the Medicane Ianos. Studies are in progress considering this point, and preliminary results show an important impact of WIVERN winds DA compared to other data sources. So, respect to this point, WIVERN plays a major role. Third we assumed that the storm is well sampled and that the Medicane is nearly at the center of the satellite swath. Considering the above limitations of this study, the results shown in this paper could represent an upper limit of what expected from the DA of WIVERN winds at the regional scale and further studies are needed to precisely quantify this point.

*Code and data availability.* The 3DVar software with the latest updates can be downloaded from the webpage [meteo.artov.isac.cnr.it](http://meteo.artov.isac.cnr.it). Data can be requested to the corresponding author.

## Appendix A: Assimilation of WIVERN Doppler by 3DVar

This Appendix provides further details about the 3DVar. We use the incremental formulation of the cost-function; let  $\mathbf{x}$  denote the state vector and  $\delta\mathbf{x} = \mathbf{x} - \mathbf{x}^b$  the increment respect to the background state vector  $\mathbf{x}^b$ , the 3DVar cost-function is:

$$J(\mathbf{x}) = \underbrace{\frac{1}{2}\delta\mathbf{x}^T \mathbf{B}^{-1} \delta\mathbf{x}}_{J^b} + \underbrace{\frac{1}{2}(\mathbf{H}\delta\mathbf{x} - \mathbf{d})^T \mathbf{R}^{-1} (\mathbf{H}\delta\mathbf{x} - \mathbf{d})}_{J^o} \quad (\text{A1})$$

where  $\mathbf{B}$  is the background error matrix,  $\mathbf{R}$  is the observations error matrix,  $\mathbf{d} = \mathbf{y}_o - H(\mathbf{x}^b)$  are the innovations and  $H$  is the forward observation operator, transforming the state vector into the observation space, and  $\mathbf{H}$  is the derivative of  $H$  respect to  $\mathbf{x}$ . The  $\mathbf{B}^{-1}$ , due to its large dimensions, cannot be calculated directly with inversion matrix techniques and we introduce a pre-conditioning transform  $\mathbf{U}$  such that  $\mathbf{B} = \mathbf{U}\mathbf{U}^T$ . With this transform the analysis control variable is  $\boldsymbol{\nu}$ , where  $\delta\mathbf{x} = \mathbf{U}\boldsymbol{\nu}$ , and  $2J^b = \boldsymbol{\nu}^T \boldsymbol{\nu}$ . The cost-function becomes:

$$J(\boldsymbol{\nu}) = \underbrace{\frac{1}{2}\boldsymbol{\nu}^T \boldsymbol{\nu}}_{J^b} + \underbrace{\frac{1}{2}(\mathbf{H}\mathbf{U}\boldsymbol{\nu} - \mathbf{d})^T \mathbf{R}^{-1} (\mathbf{H}\mathbf{U}\boldsymbol{\nu} - \mathbf{d})}_{J^o} \quad (\text{A2})$$

The transformation  $\mathbf{U}$  is given by a series of simpler transform in the  $x$ ,  $y$  and  $z$  directions, and the order of their application is important. In the 3DVar used in this work  $\mathbf{U} = \mathbf{U}_z \mathbf{U}_y \mathbf{U}_x$ , where  $\mathbf{U}_x$ ,  $\mathbf{U}_y$  and  $\mathbf{U}_z$  are computed starting from the background error matrices in the  $x$ ,  $y$  and  $z$  directions,  $\mathbf{B}_x$ ,  $\mathbf{B}_y$  and  $\mathbf{B}_z$ , respectively, using eigenvalue-eigenvector decompositions. Specifically,  $\mathbf{B}_x$  and  $\mathbf{B}_y$  are specified as correlation error matrices whose length-scales are computed from the NMC (National Meteorological Center) method (see Barker et al. (2004) for details), and the transforms  $\mathbf{U}_x$  and  $\mathbf{U}_y$  are computed by the eigenvalue-eigenvector decomposition of the  $\mathbf{B}_x$  and  $\mathbf{B}_y$ , respectively. The transform  $\mathbf{U}_z$  is calculated by the eigenvalue-eigenvector decomposition of the vertical background error matrix  $\mathbf{B}_z$  (see Federico (2013) for details). In the

600 3DVar formulation of this paper, the length-scales in the  $x$  and  $y$  directions are equal to each other and are functions of the vertical level. The length-scales are around 20-30 km from the surface up to 5 km height, then they increases from 30 km to 50-60 km in the height range 6000 m-11000 m, then the length-scales decrease again in the upper troposphere and lower stratosphere and are in the range 20-40 km for heights above 15000 m.

Figure A1 shows the simplest example of assimilating one WIVERN observation  $Y_o$  for a one-dimensional grid formed by 605 two grid-points (1 and 2) with wind components  $(U_1, V_1)$  and  $(U_2, V_2)$ , respectively. Denoting  $\xi$  the fraction distance of the observation from the grid-point 1, the vector  $H(\mathbf{x}^b)$  and the operator  $\mathbf{H}$  are given by:

$$H(\mathbf{x}^b) = \begin{pmatrix} (1 - \xi) \cos \phi \sin \theta \\ \xi \cos \phi \sin \theta \\ (1 - \xi) \sin \phi \sin \theta \\ \xi \sin \phi \sin \theta \end{pmatrix}^T \begin{pmatrix} U_1 \\ V_1 \\ U_2 \\ V_2 \end{pmatrix} \quad (A3)$$

and:

$$\mathbf{H} = \begin{pmatrix} (1 - \xi) \cos \phi \sin \theta \\ \xi \cos \phi \sin \theta \\ (1 - \xi) \sin \phi \sin \theta \\ \xi \sin \phi \sin \theta \end{pmatrix} \quad (A4)$$

Finally, the minimization of the cost-function  $J$  is done iteratively with the conjugate gradient method by calculation of the gradient  $\nabla_{\nu} J$ :

$$\nabla_{\nu} J = \nu + \mathbf{U}^T \mathbf{H}^T \mathbf{R}^{-1} (\mathbf{H} \mathbf{U} \nu - \mathbf{d}) \quad (A5)$$

**Figure A1.** The simplest example of an observation,  $Y_o$ , between two grid-points.

*Author contributions.* SF and MM coordinated the experiment; SF, RCT and CT maintain and develop the 3DVar code, SF provided the simulations; AB, CC and FM provided the WIVERN simulator; AB and MP provided feedback and information on WIVERN; SF prepared 615 the paper and the figures. All authors contributed to the revision of the paper.

*Competing interests.* The authors declare that no competing interests are present.

*Acknowledgements.* The work by CC, AB, MM, SF, RCT has been partially funded by the Italian Space Agency (ASI) project “Scientific studies for the Wind Velocity Radar Nephoscope (WIVERN) mission” (Project number: 2023-44-HH.0). The work by FM was carried out within the Space It Up project funded by the Italian Space Agency, ASI, and the Ministry of University and Research, MUR, under contract 620 n. 2024-5-E.0 - CUP n. I53D24000060005. ECMWF is acknowledged for providing IC/BC for the WRF simulations, and for providing part of the computational resources.

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
