# Peer review of "Assimilating WIVERN winds pseudo-observations in WRF model: an application to the outstanding case of the Medicane Ianos"

_EGUsphere, 2025_

## Referee Comment (RC1)

Review of **"Assimilating WIVERN winds in WRF model: an application to the outstanding case of the Medicane Ianos"** by Stefano Federico et al.

The wind velocity radar nephoscope (WIVERN) represents a pioneering spaceborne mission that aims to deliver in-cloud horizontal wind observations with fine vertical resolution. This paper explores the impact of assimilating WIVERN derived Doppler wind observations into the WRF model for the high-impact case study of medicane Ianos. Results indicated that assimilating WIVERN observations leads to a positive impact on the prediction of the medicane trajectory, reducing trajectory forecast error by 43%. It is also shown that the assimilation of such observations will improve prediction of precipitation and surface winds.

This study is innovative, interesting and well-written and I found these results of interest for the scientific community, addressing the value of next-generation wind observations in regional numerical weather prediction. However, there are several aspects that require significant clarification or improvement before the manuscript can be considered for publication. For this reason, I recommend **major revision** before it can be accepted for publication in *Weather and Climate Dynamics.*

**Major Comments:**

1. In L113-114, it is mentioned that the CTRL ensemble is initialized at 12 UTC on 16 September 2022 using the ECMWF-EPS data. First, this appears to be a typo, the correct data should be "16 September 2020", corresponding to the Medicane Ianos case. Second, it should be added information about the approximated horizontal grid resolution of the ECMWF-EPS data used for initialization, to consider the significant resolution mismatch between the coarser ECMWF-EPS fields and the 4 km grid resolution of the WRF model used in this study. More importantly, the current setup does not seem to adequately address model spin-up problem. Since the WRF model is run on a single, relatively small domain at convection-permitting resolution, the atmospheric fields initialized from a much coarser ensemble forecast would require a proper spin-up period to develop balanced mesoscale structures. However, the model begins data assimilation just three hours after initialization. This short spin-up time is likely insufficient for the model to reach a dynamically and thermodynamically consistent state at 4 km resolution, which may degrade the quality of the assimilation and forecast performance. I recommend that the authors reconsider the model spin-up strategy, either by lengthening the spin-up period or by conducting a sensitivity test to justify the chosen approach.

2. Given that the focus of this study is on the assimilation of WIVERN wind derived observations, the lack of a clear and dedicated figure showing the structure of these observations at multiple vertical levels is a significant omission. Please, add a figure that visualizes the WIVERN wind fields (i.e., direction and speed) at different pressure levels or heights. This would greatly enhance the reader's understanding of the observational coverage and characteristics. Indeed, adding a new section in the manuscript focus on the WIVERN data itself, would be very beneficial for the manuscript. It would benefit from a more-in-depth discussion of the WIVERN data

itself. For instance, how do the WIVERN winds compare with conventional wind observations, such as from radiosondes? Some sort of consistency check or statistical comparison would help assess data reliability. Also, it should be specified which vertical levels of WIVERN observations are assimilated or if there are any thinning or level selection applied. What is the lowest vertical level at which WIVERN observations are available? Understanding the vertical range of WIVERN is important, especially when discussing its impact on near-surface variables such as wind and precipitation.

3. A major concern with the current study is the design of the WIVERN$_{24h}$ assimilation experiment, which only assimilates WIVERN wind observations at 12 UTC on 17 September, when the medicane is already fully developed, followed by a 24 h free forecast until 12 UTC on 18 September. From a predictability and forecasting perspective, this approach raises questions about the broader relevance of the findings. Since the cyclone structure is already well stablished at the time of assimilation, the potential for WIVERN winds to meaningfully influence the genesis or early intensification phase is not tested. As such, the experiment does not provide insight into whether WIVERN observations can enhance forecast skill in the more critical lead-up phase, when predictability in inherently lower and guidance is more valuable for early warning. I strongly recommend that the authors design an additional experiment where WIVERN observations are assimilated prior to cyclone development, such as during the early stages on 15 September. This would allow the authors to assess whether WIVERN winds improve the characterization of the pre-convective environment, and whether that leads to improved prediction of the cyclone's formation, track, or intensity. Such an experiment would greatly increase the impact and relevance of the study by demonstrating the added value of WIVERN observations in a more operationally realistic forecasting context.

4. To improve clarity and reader understanding of the experimental set-up, add a schematic diagram or timeline that illustrates the configuration and sequencing of the different simulations performed in this study (CTRL, WIVERN$_{24h}$, WIVERN$_{3h}$). Include initialization times, assimilation windows, duration of the free forecasts, timing of observation ingestion, among others. Such a visual aid would be particularly helpful in understanding how the experiments differ in terms of when and how WIVERN observations are assimilated, and it would complement the textual descriptions in the methodology section.

5. Another aspect that needs to be improved from this study is related to the lack of standard data assimilation diagnostics in the performance of the WIVERN$_{3h}$ cycling DA experiment. Typically, in cycling DA, it is standard practice to include observation-space diagnostics to evaluate how the DA system is performing. These diagnostics often include: i) root-mean-square innovation, ii) total spread, iii) mean innovations and iv) consistency ratio, which compare the ensemble spread against the observation error (see Dowell et al., 2004; Yussouf et al., 2013 or Jones et al., 2016). These diagnostics not only help in understanding whether the system is behaving as expected but also provide confidence and robustness to the DA system. In particular, the "sawtooth" plots (e.g., Fig. 2 in Jones et al., 2016) are useful for showing the temporal evolution of these quantities across assimilation cycles, helping to assess whether the ensemble has sufficient spread and whether the assimilation is consistent with the assumed error statistics. I strongly encourage the authors to include these

diagnostics to demonstrate that the assimilation of WIVERN winds is working properly and to validate the underlying assumptions of the ensemble DA framework.

*Jones, T. A., Knopfmeier, K., Wheatley, D., Creager, G., Minnis, P., & Palikonda, R. (2016). Storm-scale data assimilation and ensemble forecasting with the NSSL experimental Warn-on-Forecast system. Part II: Combined radar and satellite data experiments. Weather and Forecasting, 31(1), 297-327.*

6. A key omission in the current manuscript is the lack of discussion regarding observation error characterization, which is essential in any data assimilation study. Specifically, the study does not clarify i) what observation error variance was assigned to the WIVERN-derived wind observations, ii) whether this error was estimated, assumed or tuned and, iii) if the same error was applied uniformly across vertical levels or observation types. Given that WIVERN represents a novel observing system, it is particularly important to justify the error assumptions used in the assimilation process. This includes specifying the source of the observation error (e.g., instrument noise, representativeness errors), and how it was implemented in the assimilation system.

7. There is a significant ambiguity in the Methodology section (L103-105) regarding the type of data assimilation approach employed. The manuscript states that a 3DVar is used, yet it also notes that the background error covariance matrix is computed from the CTRL ensemble, which is inconsistent with traditional 3DVar frameworks that typically rely on climatological, static covariance estimates. Are the authors using a pure 3DVar approach with a static covariance matrix or is this a form of ensemble-based 3DVar (En3DVar) or a hybrid 3DVar-ensemble method? If an ensemble is being used to estimate flow-dependent covariance, this must be explicitly stated and clearly explained. The choice of DA techniques is crucial to interpreting the system's ability to adjust to observational input, especially with novel data like WIVERN. Additionally, further clarification is needed regarding the NMC method (used in traditional 3DVar) mentioned for estimating background errors. What period or set of simulations was used to generate the perturbations for the NMC covariance matrix? Was any covariance inflation used? These details are essential for understanding the structure and realism of the background error covariance, which strongly affects how observations influence the analysis.

8. Results and conclusions drawn in this study are based solely on a single case study. While the results are promising and the case is certainly relevant, relying on one event significantly limits the generalizability and statistical robustness of the findings. medicanes are highly variable in structure, environment, and predictability, and it remains unclear whether the demonstrated benefits of WIVERN wind assimilation would hold across other cases with different dynamical regimes or forecast challenges. In this sense, a larger number of cases is necessary to draw robust conclusions. In addition, it would be very interesting to know how useful are the WIVERN observations when the baseline model performance (CTRL) is poor?

**Minor comments:**

The following are some suggestions that could help to improve the quality of the manuscript:

Introduction Section:

1) L25: "… *The purpose of data assimilation is …*" => "*The purpose of data assimilation (DA,* **add references***) is …*"

2) L31: "… *using a Doppler Wind Lidar.*" => Please add references.

3) L36: "*The Wind Velocity Radar Nephoscope (WIVERN) Illingworth et al. (2018); …*" => Add parenthesis before references => "*The Wind Velocity Radar Nephoscope (WIVERN)* **(***Illingworth et al. (2018); …*"

4) L45: It is introduced for the first time the acronym (DA). Remove from here (see my previous comment 1) above).

5) L45-46: It is stated that an ensemble-based data assimilation framework is used in this study. This is very confusing because later it is stated that a 3DVar, which is based on static climatological background error covariance matrix, is used. This is inconsistent. Please, clarify this point and be more specific about the DA methodology used in this study.

6) L58: Please, be more specific on the dates and hours in which medicane Ianos took place.

7) L75: After introducing acronym 3DVar, please add some references. Again, note that in L46 it was mentioned the use of ensemble-based DA method. However, here it is mentioned the use of the standard 3DVar, which is NOT an ensemble-based DA technique. Please, clarify this point carefully.

Data and Methods Section [**Section 2.1**]:

8) L79: "*WRF model 4.1 with ...*" => "*WRF model **V4.1**, with ...*"

9) L79: I suggest removing (WE) and (SN).

10) L80: Why the authors use such a reduced number of vertical levels (55)?. Add information about how vertical levels are distributed through the atmosphere (e.g., denser vertical levels near surface, …).

11) L80: "*The model horizontal resolution is ...*" => "*The model horizontal **grid** resolution is ...*"

12) L80-81: Remove "*in both WE and SN directions*"

13) L82: Remove "*with the SW and NE corners …, respectively.*"

14) L82: I suggest your 1st figure in the paper to be your numerical domain.

15) L82-85: Add a justification about the choice of these parameterizations used in this study. What are the main reasons the authors use this configuration?

16) L86: Again, at this point it is confusing why it is used the EPS if you are going to use 3DVar, which only requires a deterministic field.

17) L87: Are you using "analysis" or "forecasts" from the ECMWF-EPS? Please, add this clarification. Also, add further information about spatial resolution of the fields provided by the ECMWF.

18) L89: A direct downscaling from global model to a single domain at 4 km is used here. Why do the authors not make use of a nested approach?

Data and Methods Section [**Section 2.2**]:

19) L92: Be consistent with the notation. In the previous section, it was used 12:00 UTC, instead of 12 UTC.

20) L93: "*data assimilation  are considered*" => "*data assimilation* **experiments/simulations** *are considered*"

21) L94: What do you mean by "*a longer repetition cycle*"? Please, clarify.

22) L98: "*In the 24h cycle*" => "*In the* **24-hourly** *cycle*"

23) L98: Why do the single assimilation is performed at 12 UTC? Which was the state of Ianos at this time (initiation phase, fully developed, decaying, …)? Please add this information.

24) L99: What do you mean by "*WIVERN overpasses a mature storm system*"? It is not clear to me. Please, improve this sentence.

25) L103: Add more references to the 3DVar system. Again, this is not an ensemble-based system. If you are referring to an ensemble-based 3DVar, please specify.

26) L104: "*background error matrix*" => "*background error* **covariance** *matrix*". Replace along entire manuscript.

27) L104: Typo: "" => "**from**"

28) L105: Why the background error covariance matrix is computed at 12 UTC? Provide justification.

29) Equation 2: According with the notation, $X_{Nens}$ should be replaced by $X_{b,Nens}$

30) L113-114: The model should spin-up first using 6-12 hours. Direct downscaling from global resolution to 4 km without considering spin-up time will lead to imbalance physical fields.

31) L117: What do you mean by "*simulator*"? Do you mean "forward observation operator"? Please clarify.

32) L114: The assimilation of WIVERN observations in this study is performed in stratiform areas, where no convection is present. What do you obtained if you assimilate observations where vertical velocity W is not negligible?

33) L127: Is this assumption, right? How do you know that the medicane is well sampled by WIVERN? A two-panel figure comparing the model wind field of the medicane (left) and the observed wind field from WIVERN (right) at different vertical levels and times is missing in the manuscript and should be added.

Data and Methods Section [**Section 2.3**]:

34) L138: "*The red line in 2*" => "*The red line in* **Figure** *2*"

35) Figure 2: Remove title and add this information in the caption of the figure. Labels in x- and y- are not consistent with notation used in Fig.1. Please, use same notation.

36) Figure 3: a) Remove title of figure. Add blank space between AVG and parenthesis in y-label; b) Use same x- and y-labels as the rest of figures. Could you make larger the trajectory points. They are too small.

Results Section:

37) L197: "*has a very important impact*". Subjective comment. Please, rephrase.

38) Figure 4: Remove titles. Add blank space between Height and [cm] in y-labels.

39) Figure 5: Remove title and add this information to the caption of the figure. Consistent x- and y-labels with the rest of figures.

40) Figure 6: Use same y-axis limits for both panels. Use black colour to x- and y-labels.

41) Figure 8: Use black colour to x- and y-labels.

42) Figure 9: To small figure. Add units to panel d). Modify x- and y-labels according to the rest of figures. Use same notation.

Results Section [**Section 3.1**]:

43) Figure 10: Remove panel titles. Modify x- and y-labels according to the rest of figures. Use same notation for latitude and longitude. Add units to colorbar.

44) L289-290: What was the observed precipitation? It seems you are assessing the performance of your simulations without comparing with the observations. If not, please rephrase sentence to clarify this point.

45) L295: Add reference to Table 2, where the RMSE is shown.

46) L303: Table 2 or Table 3? Table 3 is not mentioned in the text.

47) Figure 11: Adjust the colorbar height to match the figure panels. Remove decimal places from colorbar labels.

Conclusions Section:

48) L347: "*a constellation of many (4)*". This sentence is unclear. Please clarify what the "*(4)*" stands for.

Appendix A Section:

49) L391: Should $\mathbf{B_x}$ and $\mathbf{B_y}$ been replaced by $\mathbf{U_x}$ and $\mathbf{U_y}$, respectively?

50) Equation A5: Should $\nu$, $U$ and $H$ been written in bold notation?

---

## Author Comment (AC1)

Review: "Assimilating WIVERN winds in WRF model: an application to the outstanding case of the Medicane Ianos" by Stefano Federico, Rosa Claudia Torcasio, Claudio Transerici, Mario Montopoli, Cinzia Cambiotti, Francesco Manconi, Alessandro Battaglia, and Maryam Pourshamsi

The paper simulates the expected improvement from assimilating line of sight Doppler winds from the future WIVERN mission for a medicane case study. Pseudo observations are produced by a selected member in an ensemble of 4 km WRF simulations downscaled from the ECMWF EPS. The cyclone track is found to improve in an ensemble with 3h or 24h data assimilation cycles compared to a control ensemble. The pressure, wind and precipitation are also impacted and the difference is reduced with respect to the selected member. The benefits of data assimilation are weakly affected by an increase in observational error estimate but clearly limited when using a generic instead of a cyclone specific background error.

The paper is generally interesting and presents new opportunities for the prediction of medicanes. However, it tends to be too specific and miss a broad general context, while lacking depth in the interpretation of results. In other words, how should the paper benefit to a broader community? Also, the text tends to be confusing by lacking consistency and repeating concepts and results. For these reasons, the paper needs major revisions before it can be considered for publication. General and specific comments are given below to help improve the paper quality.

We acknowledge the reviewer for the careful review of the paper and for the useful comments about this paper. We will try to put the paper in a wider and more general context and highlight the benefits to a broader community. We will answer shortly to this discussion and then we will update the paper according to the comments/suggestions in the review phase.

**General comments**

1. From the abstract and introduction, a general scientific context is missing to motivate the specific case study using specific data in a specific model configuration: what are current limitations, what will be new or different with WIVERN, what impact is expected for a cyclone, why a medicane? The scientific context should also be discussed in the conclusions, which currently lack references to previous studies

Ok. We will motivate better the specific case selected. Of course, Medicane are destructive storms and improving their prediction is of practical importance. Ianos was selected for two main reasons: a) it is among the most intense Medicanes to date; b) it was already well studied in the bibliography and comparison with other studies are possible.

The role of WIVERN observations for these storms, as well as for other storms, is expected to be very important as WIVERN will observe in cloud winds globally and at high spatial

resolution. This is the main novelty of WIVERN. Of course, there are other instruments that observe in clouds winds as sondes, AMS, aircrafts, but their spatial resolution in not comparable with WIVERN. WIVERN, with its 800 km swath, will sample from the synoptic scale to the mesoscale in a single passage.

2. There is a contradiction between the first validation of track only due to the absence of reference intensity, then validation of wind and precipitation as well but with respect to member 42 that is best in terms of track only; either include observations to assess other variables, or clarify throughout the paper what is actually achieved by data assimilation.

   The member 42 has the best agreement with the a-posteriori estimated trajectory. It is also important to stress that member 42 has a good representation of the timing when Ianos approached the Kefalonia and Zakynthos islands and that the surface pressure of member 42 is well in agreement with the observation in Palliki, as also noticed by the reviewer at the specific comment l143-144. After determining that the member 42 gives a reasonable representation of Ianos, it becomes our truth because we assimilated pseudo-observations derived from member 42 into the other members. As member 42 is our truth, the performance of the WIVERN DA on the other members is quantified by comparing these members (after DA) with member 42. This will be stated more clearly in the revised paper.

3. The interpretation of the results is somehow blurred: how does assimilating winds impact other variables relevant to the cyclone (e.g. clouds and thermodynamics) but also the steering wind of the cyclone environment to ultimately improve tracks?

   We showed that the impact of assimilating in cloud winds is transferred to the mass field, thanks to the WRF model. This has an impact on the precipitation and wind forecast, as shown in the paper. According to the comment above, we will try to extend this analysis.

4. Repetitions in the methods and inconsistent use of definitions (acronyms and symbols) throughout the paper make the read difficult

   We tried to be consistent with the use of symbols and definitions, but likely we missed some points. Some suggestions come from the specific comments below. We will review carefully the paper avoiding repetitions and inconsistent use of acronyms and symbols.

**Specific comments**

In the title is it not explicit that WIVERN has not been launched yet and pseudo observations are assimilated here

Ok. We will use the words " wind pseudo-observations" in place of "winds".

l. 4–10 This sounds like an advertisement for WIVERN and does not seem too relevant here

In these lines we state the uniqueness of WIVERN observations. We will reduce the content of the lines, nevertheless we think that it is appropriate to introduce these characteristics already at the abstract level.

l. 17 improves: reduces

Ok thanks.

l. 24 largely depends (but not only)

Ok thanks.

l. 28–29 This sentence is way too narrow in the broad context of data assimilation: wind at which levels? Forecasts at which scales? In which region, context, etc.? What about other observations?

Ok. We agree with this comment. We will add these details.

l. 36 missing (

Thanks.

l. 37 What is Earth Explorer 11?

Earth Explorer 11" (EE-11) is not a satellite that's already in orbit, but rather the name for ESA's upcoming 11th Earth Explorer mission, to be selected and developed as part of the European Space Agency's Living Planet Programme of Earth observation missions.

Earth explorer missions are proposed by the scientific community, continue to demonstrate how breakthrough technology can deliver an astounding range of scientific findings about our planet. The first mission was launched in 2009. Aeolus and EarthCare are among the EE missions.

l. 40–42 Some comments are expected for these numbers: e.g. what can be learned from such resolution compared to previous instruments such as Aeolus described above?

Of course, the high resolution and the three-dimensionality of the WIVERN observations a are very important and, considering the 800 km swath, WIVERN will fill the gap between synoptic scale and mesoscale. We will comment on this.

l. 43–44 Why? It is the first study dedicated to this specific task but what about previous studies using WIVERN or WRF (E)DA for Mediterranean storms or elsewhere?

This is simply the first study on the assimilation of WIVERN data in limited area models. Another study (Sasso et al., 2025, we will cite it) considered the problem for the global model ARPEGE. We will clarify this context.

Sasso, N., Borderies, M., Chambon, P., Berre, L., Girardot, N., Moll, P., et al. (2025) Impact of WIVERN Wind Observations on ARPEGE Numerical Weather Prediction Model Forecasts

Using an Ensemble of Data Assimilation Method. *Quarterly Journal of the Royal Meteorological Society*, e4991. Available from: https://doi.org/10.1002/qj.4991

l. 47 Here the introduction jumps from isolated and mostly technical sentences to more structured paragraphs about Mediterranean cyclones

Yes, this is to put Ianos Medicane in the context of Mediterranean cyclones. We will try to make the passage smoother, nevertheless we think that it is necessary to introduce the Mediterranean cyclones.

l. 55 missing)

Thanks.

l. 57 Why Ianos?

Ianos was chosen for two main reasons: a) it was among the most intense Medicanes; b) it is well studied. We will rephrase the sentence to consider this comment.

l. 65 Any insights about WRF from this paper?

Yes, the results of Pantillon et al. (2024) study suggest that the WRF simulation of the Ianos' trajectory were to the south of the a-posteriori estimated trajectory, as in out paper.

l. 85 Is a cumulus parametrization activated?

No, it is assumed that the convection is explicitly resolved at 4 km.

l. 94 during 24 or 48h?

During 48. Thanks.

l. 105 what about the 3h cycle?

The same as for the 24h cycle. This will be clarified.

l. 118 it is not well emphasized that the pseudo observations are given by the best member

Thanks for this comment. We will clarify it better.

l. 129 how is the scan defined for a virtual case study? Why noon rather than midnight local time?

Of course, there are infinite number of possible combinations of sampling, and we don't know at which time WIVERN will pass over the area considered in this paper. There will be storms well sampled by WIVERN, storm not sampled at all by WIVERN and storm partially observed by WIVERN (for example storm not at the center of the swath and short-lived). However, WIVERN will give a good sample for many storms during its lifetime (5 years or more) and, for these storms, a significant impact is expected, as confirmed by the results of this paper. The motivation for choosing the 12 UTC on 17 September is that: a) the storm is well formed, so we can generate a good pseudo-scene; b) the landfall is enough far (18 h before the landfall)

and the forecast can be of practical importance; c) the spread of the ensemble is in line with that of Pantillon et al. (2024).

l. 131 repetition of l. 127

Thanks. We will delete from line 127.

l. 133 CTRL using the above terminology

Thanks. We will use CTRL.

l. 135 why are there three segments between two dots?

The model output is saved every 1h and segments are every 1h. The dots are every 3h to indicate reference time. This will be clarified.

l. 137 This is not obvious from Figure 2; compute the spread?

Ok. We will report the spread.

l. 138 Figure 2

Thanks.

l. 139 From Flaounas et al. 2023?

We will add this reference.

l. 142 Why does the use of ERA5 explain the discrepancy?

We will rephrase the sentence. While ERA5 agrees well with satellite observations on the cyclone track, the ERA5 reanalysis indicates MSLP values up to 10 hPa shallower than the IFS analysis during the period of maximum intensity. However, ERA5 and IFS have a huge different horizontal resolution and this likely plays a role.

l. 143–144 This sounds like an important motivation for using the track only and should be clarified earlier

Yes thanks. This will also help to answer to general comment

l. 154–155 is this definition (6) of $\bar{D}$?

Ok. We will add $\bar{D}$ In parentheses to be clearer.

l. 161 what should be learned from Figure 3a? It is not commented apart from the two extremes

We agree. This figure will be removed maintaining the discussion.

l. 166–170 largely repeats Section 2.2

We agree. It will be deleted.

l. 171 some details on the WIVERN simulator are needed to understand this result

Ok. We will add some details on the WIVERN simulator.

l. 173 remove "which"

ok.

l. 185 should be "corrected observation error sigma$^2$_cLOS"

Right, thanks.

l. 188 how is the model error computed?

The model error is given by the square root of the diagonal elements of the vertical component of the background error matrix. A reference to Federico et al. (2013) will be added.

l. 193 clarify why 5 km (to match the sampling of WIVERN)

Yes, the settings of the simulator shown in lines 181-182 produces winds at 5 km horizontal resolution. This is one of the products provided by WIVERN.

l. 195 Start with Section 3.1?

Yes. We will introduce Section 3.1.

l. 200 which is which island on the map?

Ok. This will be clarified in Figure 5 (to be updated).

l. 200 distance $\bar{D}$?

Yes. We will say it explicitly.

l. 202 I do not get the point: the cyclone track is not directly related to the model winds, so why would only the cyclone intensity be the result of a propagation through model physics?

We apologize for not being clear. We wanted to stress that changes in the winds propagate to other parameters. We will rephrase or delete.

l. 204 compare panels b) WIV3h and a) CTRL?

Ok. We will express it more clearly.

l. 206 where is member 42 in Fig. 6b? Closer to member 42 does not necessarily means improves, as there is no reference for intensity (l. 143–144)

We will rephrase according to the comment.

l. 211 see comment on l. 129

See the answer above. Here we add that, being Ianos a long-lasting storm (few days), for sure it would have been sampled in the period 16-18 September.

l. 215 plotting member 42 in black on Fig. 2 with the other CTRL ensemble members (instead of Figure 3b) would make it easier to compare with Figs. 5 and 7 for the other experiments

Ok. We will add the member 42 in black in Fig.2, even if the trajectory of Flaounas et al. (2023) will likely be masked. We will see the output and decide the action. We would like to maintain Figure 3b to show the clear superposition between the member 42 and the best estimated a-posteriori trajectory.

l. 215–216 syntax

Thanks.

l. 216 clarify Figure 2 shows CTRL (in the text and figure caption)

Ok.

l. 219–221 This is hard to see without a time evolution of MSLP as in Fig. 6

We agree. Likely the comment will be deleted as, otherwise, we need to introduce a new figure.

l. 221–222 how changes propagate is obscure

After modifying the initial dynamic field of the WRF model through DA, the evolution of the model in the phase space changes. There is not a simple connection between what changed in the dynamic field and what obtained in the mass field. We will rephrase or delete.

l. 226 and WIV3h

Thanks.

l. 228 for WIV3h: 100 ($\bar{D}$ CTRL − $\bar{D}$ exp) / $\bar{D}$ CTRL = 76% (not 64% as in Table 1)

Sorry for the error.

l. 233 Fig 8a

ok.

l. 234 discussed below

ok.

l. 233–241 rather than discussing individual members, the member-to-member variability could be summarized by the standard deviation around the mean distance error $\bar{D}$ for each ensemble in Table 1

OK. We will shorten the discussion, nevertheless it is important to say that there is a certain variability.

l. 257 in the lower troposphere

Ok. This will be added.

l. 259 well represented compared to what?

Yes, member 42.

l. 264 why discuss the zonal wind here (vertical cross section) vs. meridional wind in the other panels (horizontal cross sections)? It is very confusing and very hard to interpret

The idea here, was to show the impact of the WIVERN DA in the analysis.

l. 265–266 how does it relate to the number of observations in Fig. 4a?

Of course there is relation with the number of observations, but it is not easy to disentangle the different contribution of first guess error.

l. 271 Why show this specific forecast time? Discussing different forecast times may help better understand how the data assimilation impacts the forecast

This specific forecast time was chosen to show that the impact of WIVERN DA is long-lasting. We assimilated 18 h before the time shown in Figure 10

l. 280 how do you know it is more realistic? (see comment on l. 206)

Close to member 42. Again, member 42 is not the truth but we assimilate pseudo-observations derived from its scene and so it becomes the truth.

l. 283 surface winds have just been discussed

The sentence refers to the surface winds in Kefalonia. We will correct it. However, considering also the comment in l.296-308 we are also evaluating to delete the analysis of surface winds in Kefalonia.

l. 285 overplotting the cyclone track would make it clearer

ok.

l. 289 clarify it is underestimated compared to member 42 (no obs here)

ok.

l. 293 "better": as above

ok.

l. 294 with respect to

ok.

l. 296–308 in the absence of obs, discussing the wind at a specific point does not look relevant: the local "error" in intensity and direction is due to the shift in cyclone track mainly, which is largely discussed already, rather than to the simulated cyclone intensity (that is higher in CTRL)

The main idea, here, was to show that the shift in the wind pattern can produce, locally, a large difference for Medicanes as they have sustained and strong winds, as noticed by the

reviewer. Considering the fact that observations are missing and that the comparison is not meaningful, it is import to stress that we assimilate winds from member 42 that becomes our truth (in real cases we assimilate observations and compare with observations). So, comparing again member 42 seems reasonable. Stated in other terms, the shift of the storm is a consequence of WIVERN winds DA and seems relevant for the case. We will clarify this in the discussion.

l. 324 please stick to the terminology defined above for the trajectory errors

Ok. Thanks for noticing the point.

l. 328 "very similar": how much is it for WIV24h?

We will add this information (34.4 km).

l. 331 what should be learned from the bias and MAE shown on Fig. 14?

The information about the MAE is redundant. We will remove it.

l. 337–341 this suggests that the NMC choice of background error matrix is not meaningful here

We would not say that NMC is not meaningful as there is a reduction of the trajectory errors compared to the CTRL experiment. Of course, it is sub-optimal compared to the background error matrix computed from the error of the day.

l. 362 Pantillon et al. (and other authors) discuss earlier initializations, while here (12 UTC on 16 September 2020) the track error is rather moderate; what would be the improvement of WIVERN data assimilation one or two days earlier?

We will try another initialization time (12 UTC on 15 September) as requested by the reviewer. Nevertheless, we cannot guarantee that the approach used in this paper could be applied for that date.

Table 1 Err(km) should be distance $\bar{D}$

Ok.

Table 2 clarify it is w.r.t. member 42

Ok.

Table 3 is not referred to in the text

Ok. We will insert the reference in the appropriate place.

Figure 1 The symbols cannot be read: please zoom in

Ok.

Figures 2, 3b, 5, 7, 9a-c Zooming in would greatly help here as well

ok.

Figure 4b using the same notations as in the text would be helpful (see equations 7–8)

Ok, we believe the reviewer refers to Figure 3b.

Figure 6 what is the background ensemble? = control ensemble CTRL?

 Yes, thanks for noticing the point.

Figure 8 what do diamonds and square represent?

We will add this information in the figure caption.

---

## Author Comment (AC2)

Review of **"Assimilating WIVERN winds in WRF model: an application to the outstanding case of the Medicane Ianos"** by Stefano Federico et al.

The wind velocity radar nephoscope (WIVERN) represents a pioneering spaceborne mission that aims to deliver in-cloud horizontal wind observations with fine vertical resolution. This paper explores the impact of assimilating WIVERN derived Doppler wind observations into the WRF model for the high-impact case study of medicane Ianos. Results indicated that assimilating WIVERN observations leads to a positive impact on the prediction of the medicane trajectory, reducing trajectory forecast error by 43%. It is also shown that the assimilation of such observations will improve prediction of precipitation and surface winds. This study is innovative, interesting and well-written and I found these results of interest for the scientific community, addressing the value of next-generation wind observations in regional numerical weather prediction. However, there are several aspects that require significant clarification or improvement before the manuscript can be considered for publication. For this reason, I recommend **major revision** before it can be accepted for publication in *Weather and Climate Dynamics*.

We acknowledge the reviewer for the careful and insightful review of the paper. We will answer shortly to this discussion and then we will update the paper according to the comments/suggestions in the review phase.

**Major Comments:**

1. In L113-114, it is mentioned that the CTRL ensemble is initialized at 12 UTC on 16 September 2022 using the ECMWF-EPS data. First, this appears to be a typo, the correct data should be "16 September 2020", corresponding to the Medicane Ianos case. Second, it should be added information about the approximated horizontal grid resolution of the ECMWF-EPS data used for initialization, to consider the significant resolution mismatch between the coarser ECMWF-EPS fields and the 4 km grid resolution of the WRF model used in this study. More importantly, the current setup does not seem to adequately address model spin-up problem. Since the WRF model is run on a single, relatively small domain at convection-permitting resolution, the atmospheric fields initialized from a much coarser ensemble forecast would require a proper spin-up period to develop balanced mesoscale structures. However, the model begins data assimilation just three hours after initialization. This short spin-up time is likely insufficient for the model to reach a dynamically and thermodynamically consistent state at 4 km resolution, which may degrade the quality of the assimilation and forecast performance. I recommend that the authors reconsider the model spin-up strategy, either by lengthening the spin-up period or by conducting a sensitivity test to justify the chosen approach.

   Yes 2022 is a typo. This will be corrected. Considering the point of the resolution mismatch between ECMWF-EPS (36 km) and our simulations (4 km) we searched for a compromise between the complexity of the model simulations and the computing time. We followed a heuristic approach, and we were guided by the comparison of our results with those reported in Pantillon et al (2024). The evolution of the trajectories of the Medicane Ianos simulated by our ensemble are compatible with those reported in this study and we used this as a proof of a reasonable setting of the model. Considering the point of assimilating WIVERN just after 3h, we agree that this could cause unbalances in the model for the $WIV_{3h}$ simulations. Nevertheless, the

simulations assimilating WIVERN every 3h smoothly converge towards the reference trajectory and, likely, unbalances are eventually mitigated by the frequent DA. However, to consider more in detail the comment raised by the reviewer, some simulations will be performed assimilating WIVERN frequently (3h) after 12h of spin-up time to better investigate this point.

2. Given that the focus of this study is on the assimilation of WIVERN wind derived observations, the lack of a clear and dedicated figure showing the structure of these observations at multiple vertical levels is a significant omission. Please, add a figure that visualizes the WIVERN wind fields (i.e., direction and speed) at different pressure levels or heights. This would greatly enhance the reader's understanding of the observational coverage and characteristics. Indeed, adding a new section in the manuscript focus on the WIVERN data itself, would be very beneficial for the manuscript. It would benefit from a more-in-depth discussion of the WIVERN data itself. For instance, how do the WIVERN winds compare with conventional wind observations, such as from radiosondes? Some sort of consistency check or statistical comparison would help assess data reliability. Also, it should be specified which vertical levels of WIVERN observations are assimilated or if there are any thinning or level selection applied. What is the lowest vertical level at which WIVERN observations are available? Understanding the vertical range of WIVERN is important, especially when discussing its impact on near-surface variables such as wind and precipitation.

Thank you for noticing this point. We will reorganize and develop more the content about the WIVERN observations. In general, WIVERN will observe the winds along the line of sight (LoS). This LoS varies in time (12 rpm) and the WIVERN observations are a combination of zonal and meridional wind components with sine/cosine of variable angles. In this sense, the WIVERN observations cannot be interpreted easily in terms of zonal and meridional wind components. We will add a figure of WIVERN winds at different levels to better show their density. Considering the error of WIVERN observations it is already full developed in the paper, as it depends on reflectivity plus other factors. In any case the final error is expected lower than 3 m/s. Of course, the error depends on the reflectivity, which is dependent on the three-dimensional structure of the observed cloud; this is, for example, much different from sondes, whose error depends on the height (the higher the height the higher the error). For the case of the Medicane Ianos we reported the WIVERN error (averaged over a single level). We will show the same curve for the wind observations of sondes. For lower levels, the sondes have a lower error; for upper level the situation is reversed. Finally, no data thinning was applied in the vertical direction, and we start to assimilate from 1 km above the surface. This limit is a bit optimistic over the land where WIVERN observations should be available starting from 2km above the surface, nevertheless we assimilate by far over the sea for the case study considered in the paper and this is not an issue.

3. A major concern with the current study is the design of the WIVERN24h assimilation experiment, which only assimilates WIVERN wind observations at 12 UTC on 17 September, when the medicane is already fully developed, followed by a 24 h free forecast until 12 UTC on 18 September. From a predictability and forecasting perspective, this approach raises questions about the broader relevance of the

findings. Since the cyclone structure is already well stablished at the time of assimilation, the potential for WIVERN winds to meaningfully influence the genesis or early intensification phase is not tested. As such, the experiment does not provide insight into whether WIVERN observations can enhance forecast skill in the more critical lead-up phase, when predictability in inherently lower and guidance is more valuable for early warning. I strongly recommend that the authors design an additional experiment where WIVERN observations are assimilated prior to cyclone development, such as during the early stages on 15 September. This would allow the authors to assess whether WIVERN winds improve the characterization of the pre- convective environment, and whether that leads to improved prediction of the cyclone's formation, track, or intensity. Such an experiment would greatly increase the impact and relevance of the study by demonstrating the added value of WIVERN observations in a more operationally realistic forecasting context.

As stated into the paper, the time for data assimilation was chosen when the storm was well formed to fulfill these two requirements: a) we are able to have at least one member that simulates well the real Ianos trajectory (this occurs, to our knowledge, only starting the forecast on the 16 September and we need this simulation for generating pseudo-observations); b) we are enough far from the landfall to ensure at least few hour of alerting time. Of course, as suggested by the reviewer, it would be interesting to investigate the WIVERN potential in the early stages of the storm, and we will design an additional experiment starting on the 15 September (12 UTC) using a similar approach of that used in the paper. Nevertheless, we cannot guarantee to have a useful member to generate pseudo-observations to fully explore the point.

4. To improve clarity and reader understanding of the experimental set-up, add a schematic diagram or timeline that illustrates the configuration and sequencing of the different simulations performed in this study (CTRL, WIVERN24h, WIVERN3h). Include initialization times, assimilation windows, duration of the free forecasts, timing of observation ingestion, among others. Such a visual aid would be particularly helpful in understanding how the experiments differ in terms of when and how WIVERN observations are assimilated, and it would complement the textual descriptions in the methodology section.
Ok. We will add this schematic diagram to be clearer in the methodology section.

5. Another aspect that needs to be improved from this study is related to the lack of standard data assimilation diagnostics in the performance of the WIVERN3h cycling DA experiment. Typically, in cycling DA, it is standard practice to include observation-space diagnostics to evaluate how the DA system is performing. These diagnostics often include: i) root-mean-square innovation, ii) total spread, iii) mean innovations and iv) consistency ratio, which compare the ensemble spread against the observation error (see Dowell et al., 2004; Yussouf et al., 2013 or Jones et al., 2016). These diagnostics not only help in understanding whether the system is behaving as expected but also provide confidence and robustness to the DA system. In particular, the "sawtooth" plots (e.g., Fig. 2 in Jones et al., 2016) are useful for showing the temporal evolution of these quantities across assimilation cycles, helping to assess whether the ensemble has sufficient spread and whether the assimilation is consistent with the assumed error statistics. I strongly encourage the authors to

include these diagnostics to demonstrate that the assimilation of WIVERN winds is working properly and to validate the underlying assumptions of the ensemble DA framework.

Thank you for the comment. We will introduce some analysis as those suggested by the reviewer. Specifically, the sawtooth plots (Figure 2 Jones et al., 2016) seems relevant for the purpose of this paper.

6. A key omission in the current manuscript is the lack of discussion regarding observation error characterization, which is essential in any data assimilation study. Specifically, the study does not clarify i) what observation error variance was assigned to the WIVERN-derived wind observations, ii) whether this error was estimated, assumed or tuned and, iii) if the same error was applied uniformly across vertical levels or observation types. Given that WIVERN represents a novel observing system, it is particularly important to justify the error assumptions used in the assimilation process. This includes specifying the source of the observation error (e.g., instrument noise, representativeness errors), and how it was implemented in the assimilation system.

We will clarify better the comments raised above. The material, however, is already into the paper. The WIVERN simulator starts from a scene of member 42. The observation errors depend on the observed signa to noise ratio (ultimately the reflectivity, see Eqn. 7), which is derived from the hydrometeors simulated by WRF member 42 at the assimilation time (the reflectivity in the W band is simulated by the WIVERN simulator starting from the hydrometeor content). Moreover, we need to add a some error to that computed in Eqn. (7) to take into account for mis pointing of the radar antenna and for partial beam filling. This is accounted in Eqn. 8. The equations (7) and (8) show the formulation of WIVERN winds along the LoS error. The error of Eqn. 8 is applied to WIVERN winds pseudo-observations along the LoS. Observations errors are assumed uncorrelated, and a data thinning of 10 km is applied (this corresponds to take one pseudo-observation every 2 pseudo-observations). The average value of the error as a function of the vertical levels is shown in Figure 4b (black curve). Incidentally, we use the term "WIVERN simulator" to indicate the forward observation operator. The name "simulator" is preferred in the radar community.

7. There is a significant ambiguity in the Methodology section (L103-105) regarding the type of data assimilation approach employed. The manuscript states that a 3DVar is used, yet it also notes that the background error covariance matrix is computed from the CTRL ensemble, which is inconsistent with traditional 3DVar frameworks that typically rely on climatological, static covariance estimates. Are the authors using a pure 3DVar approach with a static covariance matrix or is this a form of ensemble-based 3DVar (En3DVar) or a hybrid 3DVar-ensemble method? If an ensemble is being used to estimate flow-dependent covariance, this must be explicitly stated and clearly explained. The choice of DA techniques is crucial to interpreting the system's ability to adjust to observational input, especially with novel data like WIVERN. Additionally, further clarification is needed regarding the NMC method (used in traditional 3DVar) mentioned for estimating background errors. What period or set of simulations was used to generate the perturbations for the NMC covariance matrix? Was any covariance inflation used? These details are essential for understanding the structure

and realism of the background error covariance, which strongly affects how observations influence the analysis.

We are sorry for not being clear about the terminology. In the paper there are two experiments presented. The experiments WIV_12h and WIV_3h use the En3DVar approach. In our setting of the En3DVar, the background error matrix is entirely determined by the ensemble. In the experiment NMC the background error matrix is static, and it is computed for the month of September 2020 (Lines 316-323). We apologize for the confusion and we will correct this point in the revised version of the paper.

8. Results and conclusions drawn in this study are based solely on a single case study. While the results are promising and the case is certainly relevant, relying on one event significantly limits the generalizability and statistical robustness of the findings. medicanes are highly variable in structure, environment, and predictability, and it remains unclear whether the demonstrated benefits of WIVERN wind assimilation would hold across other cases with different dynamical regimes or forecast challenges. In this sense, a larger number of cases is necessary to draw robust conclusions. In addition, it would be very interesting to know how useful are the WIVERN observations when the baseline model performance (CTRL) is poor?

Ok. We will clarify better this limitation in the conclusion section. Of course, a much larger number of case studies is required to better define the usefulness/limitation of the WIVERN winds data assimilation.
* * *
*Jones, T. A., Knopfmeier, K., Wheatley, D., Creager, G., Minnis, P., & Palikonda, R. (2016). Storm-scale data assimilation and ensemble forecasting with the NSSL experimental Warn-on-Forecast system. Part II:*

*Combined radar and satellite data experiments. Weather and Forecasting, 31(1), 297-327.*

**Minor comments:**
Thank for the careful review of these minor comments. We will consider all the points and correct the paper accordingly.

The following are some suggestions that could help to improve the quality of the manuscript:
Introduction Section:

1) L25: "… *The purpose of data assimilation is* …" => "*The purpose of data assimilation (DA,* **add references***) is* …"

2) L31: "… *using a Doppler Wind Lidar.*" => Please add references.
3) L36: "*The Wind Velocity Radar Nephoscope (WIVERN) Illingworth et al. (2018); …*" => Add parenthesis before references => "*The Wind Velocity Radar Nephoscope (WIVERN)* **(***Illingworth et al. (2018); …*"

4) L45: It is introduced for the first time the acronym (DA). Remove from here (see my previous comment 1) above).

5) L45-46: It is stated that an ensemble-based data assimilation framework is used in this study. This is very confusing because later it is stated that a 3DVar, which is based on static climatological background error covariance matrix, is used. This is inconsistent. Please, clarify this point and be more specific about the DA methodology used in this study.

1. 6) L58: Please, be more specific on the dates and hours in which medicane Ianos took place.
2. 7) L75: After introducing acronym 3DVar, please add some references. Again, note that in L46 it was mentioned the use of ensemble-based DA method. However, here it is mentioned the use of the standard 3DVar, which is NOT an ensemble- based DA technique. Please, clarify this point carefully.

Data and Methods Section [**Section 2.1**]:

8. 8) L79: "*WRF model 4.1 with …*" => "*WRF model **V4.1**, with …*"
9. 9) L79: I suggest removing (WE) and (SN).
10. 10) L80: Why the authors use such a reduced number of vertical levels (55)?. Add information about how vertical levels are distributed through the atmosphere (e.g., denser vertical levels near surface, …).

11) L80: "*The model horizontal resolution is …*" => "*The model horizontal **grid** resolution is …*"

12) L80-81: Remove "*in both WE and SN directions*"

13) L82: Remove "*with the SW and NE corners …, respectively.*"

14) L82: I suggest your 1st figure in the paper to be your numerical domain.

15) L82-85: Add a justification about the choice of these parameterizations used in this study. What are the main reasons the authors use this configuration?

16) L86: Again, at this point it is confusing why it is used the EPS if you are going to use 3DVar, which only requires a deterministic field.

17) L87: Are you using "analysis" or "forecasts" from the ECMWF-EPS? Please, add this clarification. Also, add further information about spatial resolution of the fields provided by the ECMWF.

18) L89: A direct downscaling from global model to a single domain at 4 km is used here. Why do the authors not make use of a nested approach?

Data and Methods Section [**Section 2.2**]:

19) L92: Be consistent with the notation. In the previous section, it was used 12:00 UTC, instead of 12 UTC.

20) L93: "*data assimilation cycles are considered*" => "*data assimilation **experiments/simulations** are considered*"

21. 21) L94: What do you mean by "*a longer repetition cycle*"? Please, clarify.
22. 22) L98: "*In the 24h cycle*" => "*In the **24-hourly** cycle*"
23. 23) L98: Why do the single assimilation is performed at 12 UTC? Which was the state of Ianos at this time (initiation phase, fully developed, decaying, …)? Please add this information.
24. 24) L99: What do you mean by "*WIVERN overpasses a mature storm system*"? It is not clear to me. Please, improve this sentence.
25. 25) L103: Add more references to the 3DVar system. Again, this is not an ensemble-based system. If you are referring to an ensemble-based 3DVar, please specify.

26) L104: "*background error matrix*" => "*background error **covariance** matrix*". Replace along entire manuscript.

27) L104: Typo: "form" => "**from**"

28) L105: Why the background error covariance matrix is computed at 12 UTC? Provide justification.

29) Equation 2: According with the notation, XNens should be replaced by X**b,**Nens

30) L113-114: The model should spin-up first using 6-12 hours. Direct downscaling from global resolution to 4 km without considering spin-up time will lead to imbalance physical fields.

31) L117: What do you mean by "*simulator*"? Do you mean "forward observation operator"? Please clarify.

32) L114: The assimilation of WIVERN observations in this study is performed in stratiform areas, where no convection is present. What do you obtained if you assimilate observations where vertical velocity W is not negligible?

33) L127: Is this assumption, right? How do you know that the medicane is well sampled by WIVERN? A two-panel figure comparing the model wind field of the medicane (left) and the observed wind field from WIVERN (right) at different vertical levels and times is missing in the manuscript and should be added.

Data and Methods Section [**Section 2.3**]:

34) L138: "*The red line in 2*" => "*The red line in **Figure** 2*"

35) Figure 2: Remove title and add this information in the caption of the figure. Labels in x- and y- are not consistent with notation used in Fig.1. Please, use same notation.

36) Figure 3: a) Remove title of figure. Add blank space between AVG and parenthesis in y-label; b) Use same x- and y-labels as the rest of figures. Could you make larger the trajectory points. They are too small.

Results Section:

37) L197: "*has a very important impact*". Subjective comment. Please, rephrase.

38) Figure 4: Remove titles. Add blank space between Height and [cm] in y-labels.

39) Figure 5: Remove title and add this information to the caption of the figure. Consistent x- and y-labels with the rest of figures.

40) Figure 6: Use same y-axis limits for both panels. Use black colour to x- and y- labels.

41) Figure 8: Use black colour to x- and y-labels.

42) Figure 9: To small figure. Add units to panel d). Modify x- and y-labels according to the rest of figures. Use same notation.

Results Section [**Section 3.1**]:

43) Figure 10: Remove panel titles. Modify x- and y-labels according to the rest of figures. Use same notation for latitude and longitude. Add units to colorbar.

44) L289-290: What was the observed precipitation? It seems you are assessing the performance of your simulations without comparing with the observations. If not, please rephrase sentence to clarify this point.

45) L295: Add reference to Table 2, where the RMSE is shown.

46) L303: Table 2 or Table 3? Table 3 is not mentioned in the text.
47) Figure 11: Adjust the colorbar height to match the figure panels. Remove decimal places from colorbar labels.

Conclusions Section:
48) L347: "*a constellation of many (4)*". This sentence is unclear. Please clarify what the "*(4)*" stands for.

Appendix A Section:
49) L391: Should **Bx** and **By** been replaced by **Ux** and **Uy**, respectively?

50) Equation A5: Should n, *U* and *H* been written in bold notation?

---

## Author Response (AR2)

Second review of "Assimilating WIVERN wind pseudo-observations in WRF model: an application to the outstanding case of the Medicane Ianos" by Stefano Federico et al.

My previous concerns have been taken into account: the aims of the paper and the WIVERN instrument are now clearly introduced, the use of member 42 as reference has been clarified, and the interpretation of the results has improved, also with the inclusion of new contents. Overall, the paper quality has increased and the results are now more robust thanks to new sensitivity tests. However, inconsistencies remain in the methods and results, and further effort is needed before the paper can be considered for publication. Minor comments are listed below to help in that way.

First we would like to thank both reviewers for the careful review of this paper and for the useful suggestions that improved the quality of the paper.

l. 54 The synergy is not fully clear as the Aeolus mission is over

Right, but EPS-Aeolus, built on the success of the Aeolus mission, will fly with WIVERN. We added the link to the EPS-Aeolus web page https://www.eumetsat.int/eps-aeolus. We wrote:

"Although the Aeolus mission is over, the Eumetsat Polar Satellite Aeolus (https://www.eumetsat.int/eps-aeolus), built on the success of the Aeolus mission, will fly with WIVERN making the synergy possible."

l. 59 formatting issue

Corrected. Thanks.

l. 60 spell out En3DVar

Done. Thanks.

l. 67 what is the difference between ETKF and EnKF?
The ETKF (Ensemble Transform Kalman Filter) is a suboptimal Kalman filter with the forecast error covariance matrix estimated by the covariance matrix of the ensemble forecast perturbations. It is faster than EnKF and the forecast perturbations are transformed in analysis perturbation through a transformation matrix. However, the sentence was deleted as it didn't add much to the paper and going in these details is out of the scope of the paper.

Details on the ETKF can be found in:

Wang, X., and C. H. Bishop, 2003: A comparison of breeding and ensemble transform Kalman filter ensemble forecast schemes. J. Atmos. Sci., **60 ,** 1140–1158.

Wang, X., T. M. Hamill, J. S. Whitaker, and C. H. Bishop, 2007a: A comparison of hybrid ensemble transform Kalman filter–optimum interpolation and ensemble square root filter analysis schemes. Mon. Wea. Rev., **135 ,** 1055–1076.

l. 81 you may want to cite the recent paper by Miglietta et al. 2025: https://doi.org/10.1175/BAMS-D-24-0289.1
Done. Thanks.

l. 102 despite → although

Done. Thanks.

l. 135 as of September 2020 the ECMWF-ENS had horizontal resolution of ~18 km: https://confluence.ecmwf.int/display/FCST/Implementation+of+IFS+Cycle+47r1

Corrected. Sorry for the mistake.

Figure 2 typo "staring"
Corrected.

l. 161 far enough

Corrected.

l. 185 the red line in Figure 3 is hard to see; zooming in would help

Ok. In the revised version of the paper, we provided a zoom (panel b). The new figure has two panels: the Figure 3 of the previous version of the paper is panel a) and the zoom is panel b).

l. 190 as of September 2020 the ECMWF-IFS had horizontal resolution of ~9 km: https://confluence.ecmwf.int/display/FCST/Implementation+of+IFS+Cycle+47r1
Corrected. Sorry for the mistake.

Figure 4 missing time in caption and (a) (b) (c) in panels

Added.

l. 240 this sentence is awkward: (1) the vertical tilt is not shown in Fig. 1b and is a characteristic of extratropical rather than tropical cyclones; (2) the (arbitrary) swath of WIVERN is illustrated at a specific time thus cannot match the 3 hourly evolution

Yes, but Figure 1b shows the center of Ianos and the WIVERN swath used for all times. As the storm center is well inside the WIVERN swath and as the storm is in its tropical-like phase for the whole period considered, we can infer that the storm is reasonably sampled at all times. We rephrased the sentence avoiding the discussion about the tilting. We wrote:

"Figure16b) shows that the Ianos center is well inside the WIVERN swath for most of the times, so we can infer that the storm is well sampled. However, as Ianos approaches the land, its center is closer to the WIVERN swath edge and the sampling of Ianos becomes suboptimal."

l. 256 please explain

Ok. We explained better the point. We wrote:

"The observation error covariance matrix is diagonal, i.e. observation errors are assumed uncorrelated. However, observation errors are correlated, and their correlation decreases with the distance among the observations (Bennit2017, Torcasio2023). In this paper, pseudo-observations are generated at 5 km horizontal resolution and are thinned to 10 km distance to account, at least partially, for assuming a diagonal observation error covariance matrix.

l. 267 what is this radio-sounding??? it is not mentioned anywhere else in the paper

In the first review of the paper, we were asked to compare the WIVERN wind error with radio-sounding error, and we did this in Figure 5b. The radio-sounding error is that used by the WRF data assimilation software (file SOUND.txt) and it seems appropriate to use this error for comparison. We clarified better the point by rephrasing.

l. 305 reference?
We added a recent paper on this vast subject. The reference is:

Pandey, S. K. and Yadav, K.: A mathematical model for viscous flow dynamics of tropical cyclones, European Journal of Mechanics -B/Fluids, 111, 72–80, https://doi.org/https://doi.org/10.1016/j.euromechflu.2024.12.003, 2025.700

l. 307 which phase space?

We slightly changed the sentence, and the phase space is not mentioned anymore.
l. 323 far enough
Corrected. Thanks.

l. 330-331 syntax

Ok. Thank you for noticing the point.
l. 342 repetition

Thanks for noticing the point. We slightly changed the reference to Figure 10 to avoid the repetition.

372-374 as noted in my first review, Figure 11 (d) is very challenging to interpret; showing instead a zonal cross-section of the meridional wind would match the contents of panels (a-c) and be way more straightforward

Ok. We suggested to maintain our version of the figure to show the impact of WIVERN winds DA on both components of the wind, however, as suggested by the reviewer, considering only the meridional wind components is more straightforward. So, we changed the figure and the comment accordingly. The new Figure 11 is shown below:

[Figure]

Figure 11. Analysis of wind components at 12 UTC on 17 September 2020 for member 10: a) background meridional wind component at about 2500 m a.s.l; b) analysis of the meridional wind component at about 2500 m a.s.l; c) difference between analysis and background fields of the meridional wind component (same level of panels a-b); d) cross-section of the difference between the analysis and the background of the meridional wind component along the red line of panel c). The y-axis of panel d) shows the WRF vertical levels and labels on the right y-axis correspond to the approximate heights of the levels.

l. 375 despite a decrease
Done.

l. 406-408 the average precipitation over the box (and not just over Kefalonia) should be given to better understand whether the reduction in RMSE is due to an improvement in precipitation location and/or intensity

Ok. We did this calculation. Results show a minor worsening of the bias when WIVERN DA is assimilated. Specifically, the ensembles CTRL and WIV$_{24h}$ are overestimating the precipitation in the southern part of the red rectangle and underestimating the rainfall in the most intense part of the storm (i.e. north of the representative member trajectory). As WIV$_{24h}$, compared to CTRL, improves (i.e. increases) the precipitation forecast in the most intense part of the storm and tend to reduce the rainfall in the southern part of the red rectangle of Figure 14 it follows that: a) the bias is almost unchanged; b) the RMSE is notably improved. Stated in other terms, the WIVERN DA improves both the location and intensity of the precipitation forecast in the most intense part of the storm.

We wrote:

"Although the RMSE is improved by WIVERN DA, the average precipitation over the red rectangle of Figure 13 is slightly worsened. The average rainfall of the member 42, of the CTRL ensemble and of the WIV$_{24h}$ ensemble are, respectively, 59.3 mm, 63.6 mm and 64.1 mm. This result, i.e. a notable improvement of the RMSE and a similar bias, shows that the improvement of the RMSE is determined by a better representation of the location and intensity of the rainfall in the most intense part of the storm (i.e. north of the trajectory of the representative member 42)."

Figure 12 remove "eye" (surface pressure below 975 hPa has no specific physical meaning)
Deleted. Thanks for noticing the point.
Figure 13 show the orange rectangle on all panels as they are all compared over the area

Done.
l. 414-417 comparing a number of members using an arbitrary error threshold and for a specific location without showing results is not convincing; this part requires a more systematic approach (such as the RMSE applied to precipitation over a wider area above) or must be removed altogether
Ok. We deleted the comparison of the surface wind in Kefalonia for the reasons stated by the reviewer.
l. 418 referring to the previous version of the paper is not appropriate!!!
Deleted.
l. 424 what does NMC stand for?
National Meteorological Center. Added.

l. 425-428 I do not understand how the background error matrix is computed: from two forecasts (at which lead times?) or analysis/forecast, from IFS or WRF? Please clarify
The background error matrix is computed from WRF forecasts verifying at the same time. The verifying times are 00 UTC and 12 UTC for the whole month of September 2020 and the WRF

forecasts have lead times of 12h and 24h. We rephrased the sentence to clarify the point. We wrote:

"Specifically, the background error matrix was computed from the difference of two WRF forecasts, with lead times of 12 h and 24 h, verifying at the same time, both 12 UTC and 00 UTC, for the whole month of September 2020. The WRF forecasts use the operational analysis/forecast cycle issued by the ECMWF at 00 UTC and 12 UTC on each day of September as initial and boundary conditions."

l. 440 bias: lowercase

Ok.

l. 441 what is the definition of first guess here?

Thanks. It is CTRL. Modified.

l. 446 refer to fig. 14

Done.

l. 455-460 the information is repetitive: "assimilation at 12 UTC on 16 September" = "forecast from 12 UTC on 16 September" = "assimilated 24 h after the ensemble initialization" (same for WIV12h)

To avoid repetitions, we deleted the sentence "Also, in the first experiment pseudo-observations are assimilated 24 h after the ensemble initialization, while in the second experiment WIVERN pseudo-observations are assimilated after 12 h from the ensemble initialization."

l. 473 refer to fig. 16b

Thanks. Done.

l. 528 with respect to; but the sentence is awkward as the results are preliminary and not shown here

Agree. We deleted the whole sentence.

---

## Author Response (AR3)

**Public justification (visible to the public if the article is accepted and published)**:
Dear Stefano Federico and co-authors,

Thank you for the revised manuscript, which is almost ready for publication.
A last remaining issue is concerning the reviewer's comment on the use of radiosounding error in Fig. 5b. Please clarify in the text where the data is taken from.

Best regards,
Shira
* * *
Dear Shira,
tank you for your work on this paper. We added the information that we used the file SPD.txt of the WRF data assimilation package (WRFDA), and we added the ink to the package. We wrote:
"Figure 5 panel b) shows the wind observation error averaged for each vertical level (black curve), the model wind speed error (blue curve) and the radio-sounding wind errors used in the WRF data assimilation (red line) as functions of altitude. The latter was derived from the file SPD.txt available in the WRFDA tool (see https://www2.mmm.ucar.edu/wrf/users/wrfda/)."